# Alignment of Islamic Banking Sustainability Indicators with Sustainable Development Goals: Policy Recommendations for Addressing the COVID-19 Pandemic

**Amin Jan** [1,*], **Mário Nuno Mata** [2], **Pia A. Albinsson** [3], **José Moleiro Martins** [2,4], **Rusni Bt Hassan** [5] **and Pedro Neves Mata** [6,7]

1 Department of Management and Humanities, Universiti Teknologi PETRONAS, Seri Iskandar 32610, Malaysia
2 ISCAL (Lisbon Superior Institute of Accounting and Administration), Polytechnic Institute of Lisbon, 1069-035 Lisboa, Portugal; mnmata@iscal.ipl.pt (M.N.M.); zdmmartins@gmail.com (J.M.M.)
3 Department of Marketing & Supply Chain Management, Appalachian State University, Peacock Hall 4114, ASU Box 32090, Boone, NC 28608-2090, USA; albinssonpa@appstate.edu
4 Instituto Universitário de Lisboa (ISCTE-IUL), Business Research Unit (BRU-IUL), 1069-035 Lisboa, Portugal
5 IIUM Institute of Islamic Banking and Finance, International Islamic University Malaysia, Jalan Gombak, Kuala Lumpur 53000, Malaysia; hrusni@iium.edu.my
6 ESCS—Escola Superior de Comunicação Social, Lisbon Polytechnic Institute, 1549-014 Lisbon, Portugal; pmata@escs.ipl.pt
7 ISTA—University Institute of Lisbon (ISCTE-IUL), 1649-026 Lisbon, Portugal
* Correspondence: amin_jan_khan@yahoo.com or amin_17000556@utp.edu.my

**Abstract:** This study aims to establish the link of key Islamic banking sustainability indicators with the United Nations' Sustainable Development Goals (UN SDGs) as a policy recommendation for sustainable development and to mitigate the distressing impacts of the COVID-19 pandemic on the triple bottom line (people, planet, and profit). To identify the key Islamic banking sustainability indicators, the authors selected the most cited sustainability measurement indexes in Islamic banking. Initially, the indexes were divided into 10 broader themes, and then the key Islamic banking sustainability indicators were shortlisted from each theme based on their high-frequency distribution. The shortlisted sustainability indicators were then ratified to be in line with Islamic philosophy based on "Maqasid al-Shariah" (objectives of Shariah) and were subsequently grouped into the three dimensions of economic, environmental, and social sustainability based on the axial coding method. Finally, the categorized sustainability indicators were aligned with the relevant UN SDGs through the axial coding method for policy formulation, and respectively 12 propositions were developed for policy formulation. This study labeled the methodological process of this study as the ECA method (exploration, categorization, alignment). The new ECA method offers a reverse extension in the "SDG compass" developed by the Global Reporting Initiative (GRI) for aligning business policies with the UN SDGs. The process of aligning Islamic banking sustainability indicators with the UN SDGs will provide a roadmap to recovery from the COVID-19 pandemic in terms of economic, environmental, and social issues. Due to the diversity of the UN SDG framework, it covers multiples aspects for sustainable development. Therefore, considering the UN SDGs in terms of various banking instruments will mitigate the multiple distressing impacts of COVID-19 on the triple bottom line (people, planet, and profit), it will also promote a sustainable development agenda.

**Keywords:** the ECA method; UN SDGs; COVID-19 coronavirus; Islamic banking; Maqasid al-Shariah; sustainability practices; sustainability indicators; SDG governance

## 1. Introduction

COVID-19 started as a health crisis but swiftly turned into an economic crisis and is continuing to evolve into a humanitarian crisis. COVID-19 and its various impacts are

still ongoing and its full-scale consequences on the triple bottom line (e.g., people, planet, and profit) are still unknown [1,2]. According to an April 2020 International Monetary Fund (IMF) report titled "World Economic Outlook" the global economy is projected to contract by 3% in 2020 with an accumulative loss of around USD 9 trillion, which is worse than the financial crisis of 2008. According to the United Nations (UN) projections, the world is facing the worst economic recession since the Great Depression. In line with the humanitarian crisis, according to Nicola, et al. [3], the COVID-19 has affected more than one billion children in schools, which accounts for about 67% of total enrolled students globally. There are fears that due to the current turmoil students from underdeveloped or developing countries may completely lose their education. This can result in more child-labor cases. The effects of COVID-19 are incredibly diverse, i.e., from poverty to education, to health, to economic growth, etc. Therefore, one framework that can bring all these diverse aspects under one umbrella is the United Nations Sustainable Development Goals (UN SDGs). The UN SDGs have 17 goals with 169 targets and 232 indicators set to resolve diverse global problems. The framework provides an opportunity to the world for sustainable development through ensuring and considering the UN SDGs in the recovery plans.

The financial shutdown resulting from the COVID-19 pandemic globally has had a significant impact on various business sectors [3,4]. Business sectors have seen the significant scale-down of production and employment due to the reduced demand and cash flow constraints [5]. The Islamic banking industry is not isolated, it also was affected by the pandemic [6,7]. However, its measurable impact would unfold steadily. Before the pandemic, the demand for Islamic banking financing, deposits, and assets grew gradually. According to the Islamic Financial Services Industry Stability Report 2019, the cumulative annual growth rate (CAGR) of Islamic banking for the period of 2013–2018 increased by 7.1% for Islamic finance, 7.4% for deposits, and 7.2% for Islamic banking assets. Now, in the context of the imminent global recession, the Islamic banking industry, along with the rest of the world, is looking forward to a long path of recovery and confronting new business challenges.

Extant literature suggests that Islamic banks showed a paucity of compliance towards adopting the UN SDGs in terms of sustainable business practices [8–21]. This lack of compliance towards the UN SDGs may increase regulatory challenges and stakeholders' pressure for Islamic banks at different levels. The improvement in education has equipped multiple stakeholders for global business trends, including the positive effect of sustainable business practices on Islamic banks' financial results. A huge body of knowledge is available on the subject that indicates that sustainability practices improve financial performance in the case of Islamic banks [19,22]. In this case, the investors and depositors of Islamic banks would demand their banks to be prudently involved in sustainability practices, because they will get an economic return against it. Other stakeholders in the form of different public groups and interest groups that are advocating for the environment are also putting pressure on banks to comply with the environmentally friendly activities and practices. The lack of compliance is also increasing regulatory challenges for the Islamic banking industry in different countries. For instance, Bank Negara Malaysia (BNM), through its policy paper called "Value-Based Intermediation" (VBI), nudged Islamic banks to commence sustainable business practices [23]. Therefore, to pacify multiple stakeholders and regulators, the adoption of sustainable business practices has almost become mandatory for Islamic banks.

The COVID-19 outbreak not only highlighted the urgency of research into a vaccine but also shifted the dynamics of Islamic banking towards new business trends such as clarity and solutions for a sustainable Islamic banking industry. Haider Syed, et al. [4] and the United Nations Development Program (UNDP) in its recent report (https://www.undp.org/content/undp/en/home/blog/2020/islamic-finance-takes-on-covid-19.html) published in April 2020 highlighted that Islamic sustainability instruments (such as zakat and Qard-e-Hassan) can help nations in preparing for, responding to, and recovering from the pandemic [24]. This provides an opportunity for Islamic banking to help various stakeholders suffering through the pandemic [25]. Against that background, this article

aims to identify the key Islamic banking sustainability indicators and align them with the SDGs. Firstly, it will unleash the potential of Islamic banking sustainability indicators to various stakeholders around the world. Secondly, it will provide policy guidelines for the stakeholders in responding to the pandemic using specific business instruments. Thirdly, and most importantly, it will help the world to recover from the pandemic in terms of economic, environmental, and social business issues. Finally, these fundamental steps will pave the way for sustainable Islamic banking in the long run. The following sections show the literature review and theoretical framework, followed by the linkage of Islamic banking sustainability indicators with the UN SDGs and the commentary. Last, this article provides a conclusion, future directions contribution, and policy guidelines.

## 2. Literature Review

### 2.1. The Concept of Islamic Banking and its Global Profile

Islamic banking refers to a banking system based on the laws of Islam known as Shariah objectives, or Maqasid al-Shariah, and guided by Islamic economics [19]. The two basic principles that distinguish Islamic banking from their conventional counterparts are sharing of profit and losses [26], and prohibition of interest [27]. Islamic banking is the main component of Islamic finance other than Sukuk (Islamic bonds), Islamic funds, and Takaful (Islamic insurance). The breakdown of global Islamic finance is shown in Figure 1 below.

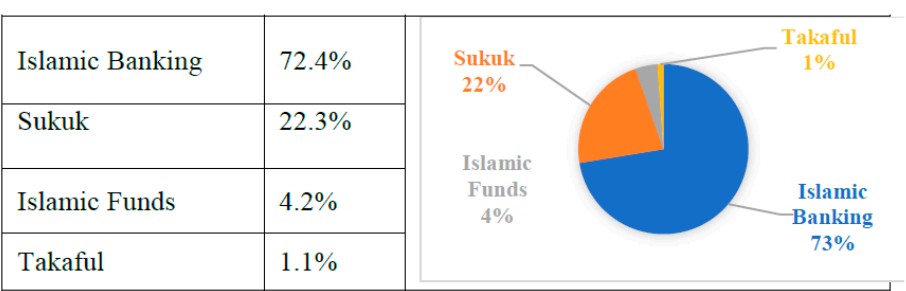

| | |
|---|---|
| Islamic Banking | 72.4% |
| Sukuk | 22.3% |
| Islamic Funds | 4.2% |
| Takaful | 1.1% |

**Figure 1.** Breakdown of Islamic finance (year-end 2019) (Source: Islamic Financial Services Industry Stability Report 2020, p. 13).

The above Figure 1 shows the data as per the year-end 2019. The Islamic banking industry retains the major portion of the global Islamic financial market. According to the Islamic Financial Services Industry Stability Report 2019, the share of global Islamic banking assets is about USD 1.57 trillion. The subsequent Figure 2 shows the share of Islamic banking assets per country.

Figure 2 shows data as per the third quarter of 2019. The market leaders in the world of Islamic banking are Iran, Saudi Arabia, Malaysia, UAE, Kuwait and Qatar. These six countries collectively possess approximately 85 percent of the world Islamic banking assets share. The second tire Islamic banking countries as per global Islamic banking shares are Turkey, Indonesia, Bangladesh and Pakistan. While the countries with the least share in the world Islamic banking industry are Brunei, Egypt, Oman, Bahrain, and Sudan.

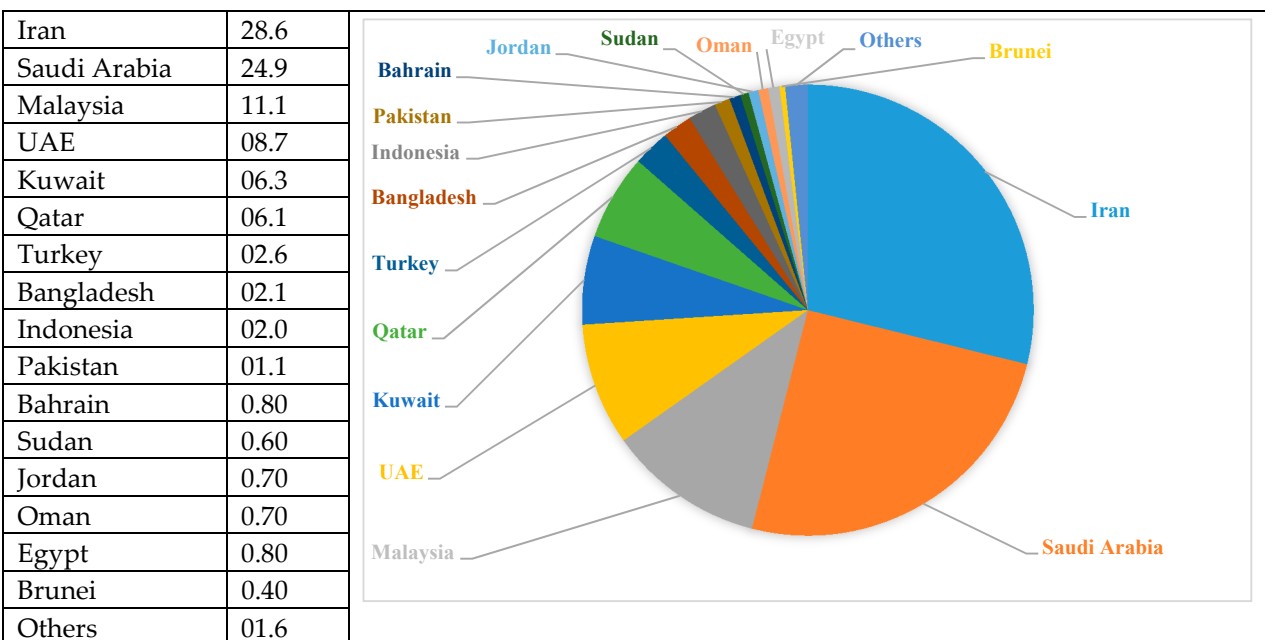

| | |
|---|---|
| Iran | 28.6 |
| Saudi Arabia | 24.9 |
| Malaysia | 11.1 |
| UAE | 08.7 |
| Kuwait | 06.3 |
| Qatar | 06.1 |
| Turkey | 02.6 |
| Bangladesh | 02.1 |
| Indonesia | 02.0 |
| Pakistan | 01.1 |
| Bahrain | 0.80 |
| Sudan | 0.60 |
| Jordan | 0.70 |
| Oman | 0.70 |
| Egypt | 0.80 |
| Brunei | 0.40 |
| Others | 01.6 |

**Figure 2.** Domicile of Islamic banking assets per country: 3Q 2019 (in percentage) (Islamic Financial Services Industry Stability Report 2020, p. 15).

### 2.2. What Are the Objectives of Shariah (Maqasid al-Shariah)?

Maqasid al-Shariah refers to the highest objectives of Islamic laws set out to achieve socioeconomic justice [28]. It has three main components, namely, necessities/essential elements, complementary elements, and embellishments. According to the theory, necessities are essential elements of human life, the absence of which may cause damage or harm to human life. Examples are food, clothes, and shelter. The necessities are further sub-divided into five elements of preservation (See Figure 3). Shariah rulings aim primarily to protect these five elements from any harm. The complimentary elements are those items that complement the necessary. This relates to the fact that the negligence of such elements does not lead to the destruction of society, but rather to certain social suffering. In other words, they are required to relieve society's hardships. Examples of complimentary items are marriage, healthy food, communication tools, and means of transportation, among others. The embellishments are not mandatory in Shariah but contribute to the perfection of society if they are performed. Examples are charitable and philanthropic work. The concept is shown in Figure 3 below.

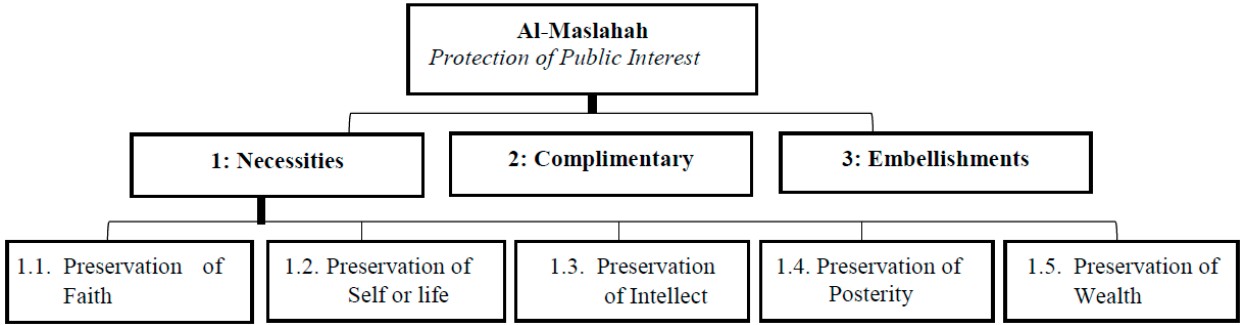

**Figure 3.** Maqasid al-Shariah theoretical framework of Al-Ghazali [28].

### 2.3. The United Nations Sustainable Development Goals (UN SDGs)

The United Nations Sustainable Development Goals (SDGs) are the collection of 17 goals with 169 targets and 232 indicators set to resolve diverse problems in the world in a sustainable manner [29]. The UN SDGs were launched in 2015 and are intended to be achieved by 2030. The UN SDGs mainly focus on achieving the five Ps, i.e., people, planet, prosperity, peace, and partnership [30]. The UN SDGs remains the area of pursuit for many researchers from the field of business sustainability [31]. The UN SDGs are shown in the table below.

The above Table 1 shows 17 SDGs. As the philosophy of the SDGs is based on the Triple Bottom Line (TBL) approach of sustainability. Therefore, in inconsistency with [29], this study categorized these goals into the three dimensions as below.

**Table 1.** United Nations Sustainable Development Goals.

| Goal: 01 | No Poverty | Goal: 10 | Reduce Inequalities |
|---|---|---|---|
| Goal: 02 | Zero Hunger | Goal:11 | Sustainable Cities and Communities |
| Goal: 03 | Good Health | Goal: 12 | Responsible Consumption |
| Goal: 04 | Quality Education | Goal: 13 | Climate Action |
| Goal: 05 | Gender Equality | Goal: 14 | Life Below Water |
| Goal: 06 | Clean Water and Sanitation | Goal: 15 | Life on Land |
| Goal: 07 | Renewable Energy | Goal:16 | Peace and Justice |
| Goal: 08 | Good Jobs and Economic Growth | Goal: 17 | Partnership for the Goals |
| Goal: 09 | Innovation and Infrastructure | | |

Table 2 shows the categorization of the 17 SDGs as per the triple bottom line. It shows that 12 SDGs purely dominate in separate sustainability dimensions, whereas the remaining 5 SDGs are interconnected, as per the Triple Bottom Line TBL.

**Table 2.** Categorization of the 17 SDGs into the triple bottom line (TBL).

| Relevance | Economic Sustainability | Environmental Sustainability | Social Sustainability |
|---|---|---|---|
| Pure Dominance | • Goal 8 <br> • Goal 9 <br> • Goal 10 | • Goal 6 <br> • Goal 7 <br> • Goal 13 <br> • Goal 14 <br> • Goal 15 | • Goal 1 <br> • Goal 4 <br> • Goal 5 <br> • Goal 16 |
| Interconnected Goals | • Goal 2 <br> • Goal 3 <br> • Goal 17 | • Goal 11 <br> • Goal 12 <br> • Goal 17 | • Goal 2 <br> • Goal 3 <br> • Goal 11 <br> • Goal 12 <br> • Goal 17 |

### 2.4. Theoretical Foundation

The below Figure 4 illustrates the proposed alignment of Islamic banking sustainability indicators with the UN SDGs based on Maqasid al-Shariah (objectives of Shariah). It confirms whether a sustainability indicator is in line with the objectives of Shariah or not. It further suggests in which category of Shariah the sustainability indicators fall. It identifies whether the sustainability indicators are related to the necessities category of Shariah objectives (refer Figure 3), or whether they relate to the complementary or embellishment components of the Shariah objectives. The alignment of Islamic banking sustainability indicators with the UN SDGs is possible only when the sustainability indicators are confirmed to be in line with the objectives of Shariah. Therefore, the alignment of Islamic

banking sustainability indicators with the UN SDGs is predominantly dependent on the endorsement of Maqasid al-Shariah (objectives of Shariah).

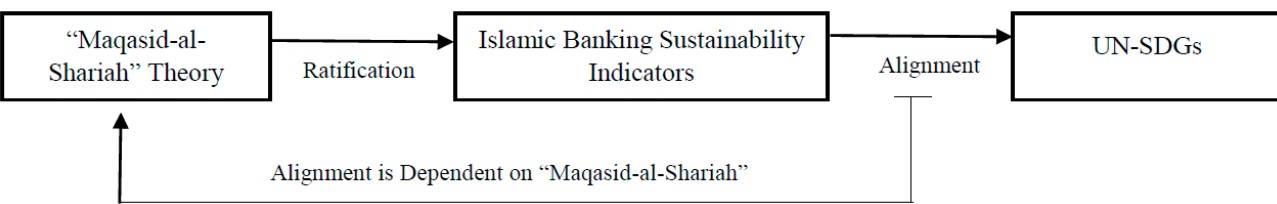

**Figure 4.** The nexus of Maqasid al-Shariah with the Sustainable Development Goals (SDG)s.

### 3. Methodology

#### 3.1. Grounded Theory

This study is based on the grounded theory concept. The theory was presented by [32]. Grounded theory is a systematic methodology in the social sciences involving the construction of new theories and new concepts through methodical gathering and analysis of data. Although initially based on what had already been learned, the grounded theory allows for further exploration to provide a deeper understanding of a subject. The grounded theory does not specify any particular type of research strategy, data, or specific theoretical foundations to develop concepts [33,34]. The grounded theory is based on three elements: concepts, categories, and propositions [35–37]. The theory gradually emerged from the data: from data to codes, from codes to concepts, from the concepts to categories, and from categories to new theories, frameworks, or indexes. According to [32], axial coding is the obligatory element of the grounded theory, and if those coding paradigms are not followed, the proposed methods/concepts/framework provide inaccurate precision [34]. In the same vein, this study explains open coding and axial coding below.

#### 3.2. Open Coding and Axial Coding Methods

According to [32], open coding deals with posting sanitizing questions related to the concept and categories of the study. According to [32], the generation of questions is based on the inductive knowledge of the researcher learned from the literature or experience in the field. Based on the list of questions offered by [32,34,38,39], for an efficient interpretation of the data this study constructed four questions related to each factor as per axial coding (see Tables A3 and A4 Appendix A). The second coding method is the axial coding method (ACM), which deals with making connections between different concepts and categories in the context of the grounded theory and designed questions. Reference [32] proposed the paradigm of axial coding based on the following factors, i.e., phenomena (context and intervening condition), causal condition, strategies, and consequences [34]. Though in the fourth edition of his book [38], he reduced the number of factors in the coding paradigm to three, i.e., action–interaction, conditions, and consequences. This study, however, to capture a holistic picture, establishes the connection between the concepts and categories based on the earlier factor of four [32]. Academic professors and industrial experts were involved in the overall creative process of open coding and axial coding.

#### 3.3. The New ECA Method

To align the key sustainability indicators of the Islamic banks with the UN SDGs, this study proposes the new ECA method, the process for which is explained in detail below.

#### 3.3.1. Step 1: Exploration (E)

This study anticipates that to align the indicators, instruments, or policies of any business (Islamic banks in this case) with the UN SDGs it is essential to first explore the pivotal indicators, instruments, or policies of the business using a systemic methodological process. For this purpose, methodologies such as principal component analysis (PCA), exploratory factor analysis (EFA), or frequency distribution, etc., can be used. These

methods will help the business firms identify their desired key indicators, instruments, or policies for alignment with the UN SDGs.

### 3.3.2. Step 2: Categorization (C)

In the second step, this study anticipates that the explored indicators, instruments, or policies must be categorized into broader dimensions/groups to make sure that they are in line with the existing business groups/dimensions. The categorization of the explored phenomena into different dimensions/groups will make the alignment easier and more rational. The group-wise alignment will allow the personnel from the group/dimension to work exclusively towards their identified SDGs. To categorize the explored business phenomena into different groups the axial coding method (ACM) must be used. The ACM is the process of the grounded theory used for establishing meaningful linkages between the categories (refer to Section 3.2 above).

### 3.3.3. Step 3: Alignment (A)

After the detailed exploration and categorization, the next step is to align the potential indicators, instruments, or policies with the respective UN SDGs. The alignment process brings synchronization between the groups/dimensions and the respective UN SDGs. The alignment process clarifies the role of authorities from different groups/dimensions towards policy formulation. For instance, the alignment process will clarify the role of economic, environmental, and social stakeholders of the firms to work explicitly towards the UN SDGs that are related to their department. In a way, the alignment process makes the compliance of business phenomena with the UN SDGs more specific. The process of alignment is subjected to the four factors of the axial coding paradigm. It implies that the process of alignment would be valid only for the present situation. In a future situation, the firms would require a fresh alignment. This is because the consequences, causal conditions, and phenomena can vary from time to time [34]. Therefore, keeping in mind the possible change to any of the four factors of the axial coding paradigm, it is prudent to conduct a fresh alignment in the future. Based on that, the firm may carry or drop certain past alignments based on the change in the consequences, causal conditions, or phenomena. To keep a check on the evolving business phenomena, causal conditions, and consequences, this study proposes a new SDG governance framework (Figure 7) that accounts for any change in the axial coding paradigm and acts based on the continuation or dropping of the alignment.

The detailed methodological process of this study is shown in Figure 5 below.

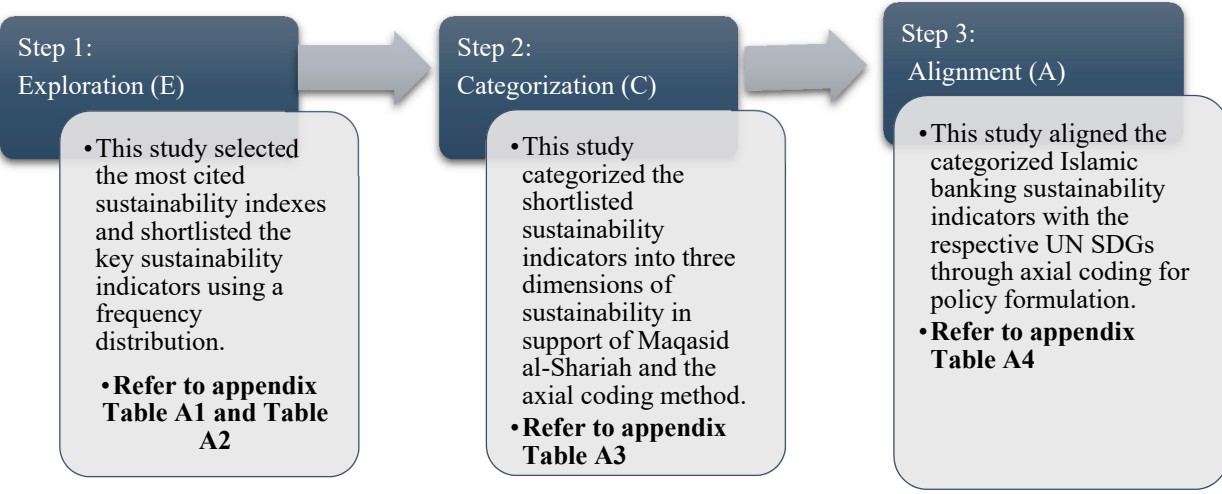

**Figure 5.** Methodological flow chart: the proposed ECA method.

### 3.4. Methodological Flow Chart

Figure 6 shows the detailed process of aligning vital business strategies (sustainability indicators in this case) with the SDGs. The SDG -Compass proposed by the Global Reporting Initiative (GRI) offers a framework for aligning business policies with the SGDs. However, the SDG -Compass is criticized on the ground that it lacks focus for exploring vital organizational goals. It only focuses on the implementation phase of the SDGs [39]. Emerging industries such as Islamic banking require detailed prior methodological knowledge before moving to the aligning phase. In the same vein, this study, by proposing antecedents (step 1 and step 2) to the SDG Compass, offers a reverse extension in the method. It will assist the Islamic banking industry in a more systematic way to align their vital business strategies with the SDGs.

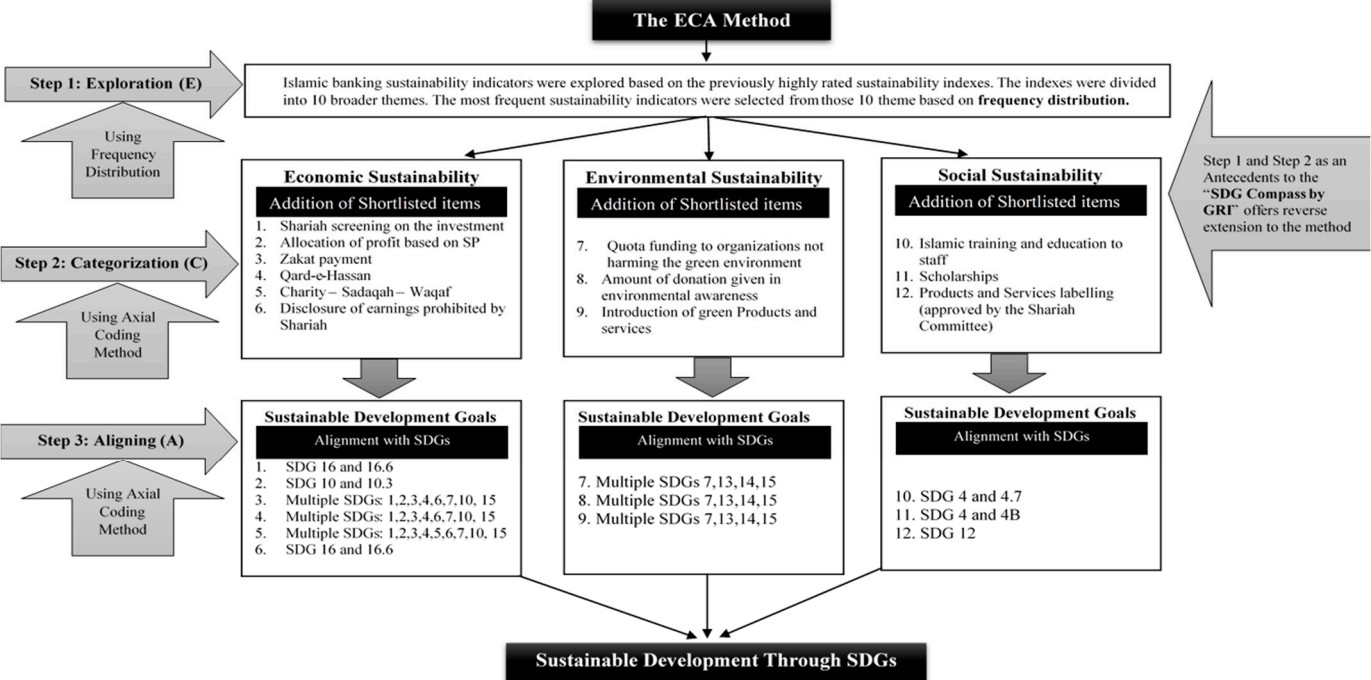

**Figure 6.** Flow chart of the proposed ECA method.

The above Table 3 shows the linkage of Islamic banks' sustainability indicators with the UN SDGs in support of Maqasid al-Shariah. The linkage of each Islamic bank's sustainability item is built with the appropriate UN SDGs (including the main goals and sub-targets). The following section shows commentary on each item in three different steps. In the first step, it shows how the sustainability indicator is categorized into the three dimensions of sustainability, i.e., economic, environmental, and social, based on the Maqasid al-Shariah theory. In the second step, it shows the alignment of the sustainability indicators with the UN SDGs. In the third step, it shows the policy implication to deal with the COVID-19 pandemic. Hence, the policy-based novelty of each sustainability indicator under this section is threefold, i.e., categorization (alpha), alignment (beta), and the COVID-19 responses based on alpha and beta.

**Table 3.** Categorization and the subsequent alignment of Islamic banking sustainability indicators with the UN SDGs.

| Economic Sustainability Indicators (Profit) | | |
|---|---|---|
| **Sustainability Indicators** | **Linkage with Maqasid al-Shariah** | **Alignment with the Sustainable Development Goals (SDGs)** |
| 1. Shariah screening of investments | Necessities—preservation of faith | SDG 16 and sub-goal 16.6 |
| 2. Allocation of profit based on Shariah principles | Necessities—preservation of faith and wealth | SDG 10 and sub-goal 10.3 |
| 3. Zakat payment | Necessities—preservation of faith and wealth | SDGs 1, 2, 3, 4, 6, 7, 10, 15 |
| 4. Qard-e-Hassan | Complementary—preservation of self or life and posterity | SDGs 1, 2, 3, 4, 6, 7, 10, 15 |
| 5. Charity—Sadaqah—Waqaf | Embellishment | SDGs 1, 2, 3, 4 |
| 6. Disclosure of earnings prohibited by Shariah | Complementary—preservation of faith, self, and wealth | SDG 16 and sub-goal 16.6 |
| **Environmental Sustainability Indicators (Planet)** | | |
| **Sustainability Indicators** | **Linkage with Maqasid al-Shariah** | **Alignment with the Sustainable Development Goals** |
| 7. Funding for organizations upholding green environment | Necessities—preservation of posterity and preservation of life | SDGs 7, 13, 14, 15 |
| 8. Amount of donations to environmental awareness | Necessities—preservation of posterity and preservation of life | SDGs 7, 13, 14, 15 |
| 9. Introduction of green products and service | Necessities—preservation of posterity and preservation of life | SDGs 7, 13, 14, 15 |
| **Social Sustainability Indicators (People)** | | |
| **Sustainability Indicators** | **Linkage with Maqasid al-Shariah** | **Alignment with the Sustainable Development Goals** |
| 10. Islamic training and education for the staff | Complementary—preservation of intellect | SDG 4 and sub-goal 4.7 |
| 11. Offering scholarships | Complementary—preservation of intellect | SDG 4 and sub-goal 4B |
| 12. Approval of new products and services by the Shariah committee | Necessities—preservation of faith | SDG 12 |

### 3.5. Economic Sustainability Indicators

The categorization and subsequent alignment of the shortlisted Islamic banking economic sustainability indicators are consistent with the doughnut economic model [40]. The doughnut framework suggests considering the efficiency of an economy to the degree that people's needs are fulfilled without overshooting the ecological ceiling of the world. Both the doughnut and the SDGs are related to sustainable development. However, the scope of the SDGs is much wider compared to the doughnut economic model.

3.5.1. Shariah Screening of Investments

■    *Categorization of Sustainability Indicator into Respective Sustainability Dimension (Alpha)*

Based on the principles of Maqasid al-Shariah, this article suggests that the instrument of "Shariah screening of investments" falls under the category of necessities and the subcategory of "preservation of faith." Islam forbids investments in the haram business (forbidden by Islamic laws), such as gambling. This is because in such trade practices the element of free and fair exchange of goods and services is not observed, and rather are based on deceit and dishonesty [41]. Therefore, it is the responsibility of the Shariah committee to screen all investment of the Islamic banks for the preservation of faith. Shariah screening of investments is categorized in the economic sustainability dimension based on the philosophy that once the Shariah committee certifies an investment of the Islamic bank

is in line with objectives of Shariah, it attracts more investors, and holistically it positively affects the economic sustainability of Islamic banks.

■ *Alignment of Sustainability Indicator with the UN SDGs (Beta)*

The instrument of Shariah screening of investments may serve SDG 16, i.e., peace, justice, and strong institutions, and its sub-target 16.6, which is about developing transparent, accountable, and effective institutions at all levels. Shariah screening of investments offers great transparency and accountability based on Islamic laws in developing effective institutions. Shariah objectives aim to promote social welfare (Al-Maslahah); therefore, the instrument of Shariah screening of investments will ensure the prevention of investments in the inappropriate haram business (forbidden by Islamic laws), such as gambling, which generally violates the business objectives of free and fair exchange.

■ *The COVID-19 Response: Based on Alpha and Beta*

The current scenario demands fair investment (Shariah screening) and strong institutions to fight the pandemic. [42] alluded that Shariah screening of investments provided hedging benefits to various market stakeholders during the pandemic. Providing such benefits during the pandemic shows the unique quality of a financial system in a time of crisis. Against that background, it shows that Shariah screening of investments helped build strong institutions during the pandemic. Hence, the following proposition was developed.

**Proposition 1.** *The alignment of Shariah screening of investments with UN SDG 16 and sub-target 16.6 will help build strong institutions to fight the COVID-19 pandemic.*

3.5.2. Allocation of Profit Based on Shariah Principles

■ *Categorization of Sustainability Indicator into Respective Sustainability Dimension (Alpha)*

The article claims, based on the principles of Maqasid al-Shariah, that the allocation of profit based on Shariah principles falls within the main category of necessities and the subcategories of faith and wealth preservation. Distribution of profit in Islamic banks is directly linked to religiosity, which directs Islamic banks to equally distribute income and minimize inequality. In a way, it preserves the wealth of stakeholders and thereby is incorporated into the economic dimension of sustainability.

■ *Alignment of Sustainability Indicator with the UN SDGs (Beta)*

This study made an alignment of the instruments with UN SDG-10 and sub-goal 10.3, which alludes to reducing inequalities of outcomes by eradicating discriminatory principles and laws. The Shariah objectives/laws aim to promote socioeconomic justice [28]. Therefore, the allocation of profit based on Shariah principles will eradicate discriminatory principles by promoting the fair distribution of profit and hence strongly supports UN SDG 10.3.

■ *The COVID-19 Response: Based on Alpha and Beta*

Profit-sharing in Islamic banks is different than conventional banks, i.e., in Islamic banks the share of profit increases only once the bank makes a profit, whereas in conventional banks the customers get a fixed rate even if the bank is not earning likewise in the pandemic [43]. Furthermore, Islamic banking clients get a high profit once the bank makes more profit, whereas in conventional banks the clients do not receive any additional share of the profit even when the bank makes more profit [43]. This situation is reducing the inequalities of outcomes for the Islamic banking clients. Even in times of crisis, unlike conventional banks, Islamic equity funds provide hedging benefits to various market stakeholders, such as during the pandemic. Times of crisis can be used as an opportunity by Islamic banks to invest in policy formulation for reducing inequalities [25]. Islamic banks are offering a practical solution to it, hence, it can be idealized by other businesses to

reduce inequalities and promote fair principles for profit-sharing during the COVID-19 pandemic. Hence, the following proposition was developed.

**Proposition 2.** *Aligning the allocation of profit based on Shariah principles with UN SDG 10 and sub-target 10.3 will help in fighting the COVID-19 pandemic by reducing inequalities of outcomes through eradicating discriminatory principles and laws.*

3.5.3. Zakat Payment

■  *Categorization of Sustainability Indicator into Respective Sustainability Dimension (Alpha)*

In Islam, zakat is a religious obligation or tax treated as the second highest in rank after prayers. It refers to a religious obligation where an individual is required to pay at the ratio of 2.5% above the minimum amount of savings per year as almsgiving [44]. This article posits that according to the principles of Maqasid al-Shariah, the zakat payment instrument falls under the main category of necessities and the subcategories of faith preservation and wealth preservation. This is because paying zakat is one of the five essential pillars of Islam. Furthermore, according to Islamic literature, it purifies and increases wealth. It is categorized into an economic dimension of sustainability as it boosts income. The payment of zakat and its transparent reporting, from the perspective of Islamic banking shareholders, would increase the goodwill of Islamic banks towards their stakeholders. As a result, the bank will obtain more funds from various stakeholders to enhance Islamic banks' economic viability in a positive direction.

■  *Alignment of Sustainability Indicator with the UN SDGs (Beta)*

Zakat payment is a unique sustainability instrument in Islamic banking that may serve multiple UN SDGs. Zakat is paid with the aim of supporting the disadvantaged people in society [45]. In the same vein, this study aligned the instrument of zakat to UN SDGs 1, 2, 3, 4, 6, 7, 10, and 15 because these goals are directly related to the needs of disadvantaged people. Hence, aligning and channeling zakat with the goals ensure compliance of the Islamic banks with the UN SDGs.

■  *The COVID-19 Response*

Haider Syed, et al. [4], also argued that zakat payment can be used as a tool for poverty alleviation during the COVID-19 pandemic. Islamic banks can design a portfolio by channeling zakat to support the disadvantaged sector of society. For instance, the pandemic has directly affected the food, poverty, health, drinking water, and education of multiple stakeholders. The instrument of zakat can be channeled into the following SDGs: goal 1 (no poverty), 2 (zero hunger), 3 (good health), 4 (quality education), 6 (clean water and sanitation), 7 (renewable energy), 8 (good jobs and economic growth), 10 (reducing inequalities), and 15 (life on land) for sustainable development. The Islamic bank may design a diversified portfolio with the help of a micro-examination to identify which area of the country/region/world requires which kind of assistance and what is most urgently required, whether it be food, medicines, drinking water, energy, or jobs, and to channel zakat to the neediest areas accordingly. In the way, it will serve multiple UN SDGs and will promote sustainable development across the country/region/world. Based on the discussion the following proposition was developed.

**Proposition 3.** *Channeling the zakat payment to UN SDGs 1, 2, 3, 4, 6, 7, 10, and 15 can assist in poverty alleviation, reducing hunger, promoting health care, offering quality education, providing clean water and sanitation, facilitating renewable energy, protecting good jobs and economic growth, reducing inequalities, and uplifting life on land during the distressing impacts of the COVID-19 pandemic.*

### 3.5.4. Qard-e-Hassan (Benevolent Loans)

■    *Categorization of Sustainability Indicator into Respective Sustainability Dimension (Alpha)*

Qard-e-Hassan is defined as the interest-free loans granted to persons in need for a specific period [46]. This study classified Qard-e-Hassan as a complementary item to the subcategories of preservation of self or life and preservation of posterity based on the principles of Maqasid al-Shariah. The Qard-e-Hassan financing facility is more applicable in poor or underdeveloped countries where it can serve to remove hardship from society and life. It is incorporated into the economic dimension of sustainability because the provision of Qard-e-Hassan to poor customers would not only increase the goodwill of Islamic banks in the minds of their customers but also of the general public. As a result, Islamic banks will attract more deposits from other stakeholders, which may strengthen their economic sustainability positively.

■    *Alignment of Sustainability Indicator with the UN SDGs (Beta)*

The instrument of Qard-e-Hassan (interest-free loans) can also target multiple UN SDGs. This loan aims to assist the needs of stakeholders for a shorter period. In line with that, this study aligned the Islamic banking sustainability instrument of Qard-e-Hassan with UN SDGs 1, 2, 3, 4, 6, 7, 10, and 15. These goals are interconnected with each other. Improvement in one goal, for instance no poverty, as a subset brings an improvement in the other goals as well. For example, alleviating poverty enhances the purchasing power of stakeholders, which allows them to afford food, health services, water, and energy. Against that background, this study posits an alignment between Qard-e-Hassan with multiple UN SDGs.

■    *The COVID-19 Response: Based on Alpha and Beta*

We posit that the payment of Qard-e-Hassan to skilled employees/entrepreneurs of Small and Medium-sized Enterprises SMEs who lost their jobs/business during the pandemic will bring positive change in the other interconnected UN SDGs. For a short period, Islamic banks need to increase the amount of Qard-e-Hassan mostly to these skilled workers. This is because they are most likely to generate economic activity with less training and effort required, because they are already trained enough and have experience in running certain small enterprises. At the same time, based on their business expertise they are most likely to repay the loan to the Islamic banks within the specified period. Haider Syed, et al. [4] alluded that Qard-e-Hassan can be used to alleviate poverty (SDG 1) during the pandemic. Poverty alleviation increases purchasing power, which as a subset allows a stakeholder to purchase better health services, buy food and drinking water, and afford electricity, etc. Hence, the following proposition was developed.

**Proposition 4.** *The payment of Qard-e-Hassan to skilled workers can lead to poverty alleviation; reduce hunger; open up access to good health care, quality education, clean water and sanitation, and renewable energy; reduce inequalities; and uplift life on land during the COVID-19 pandemic.*

### 3.5.5. Donating to Charity through Sadaqah and Waqf

■    *Categorization of Sustainability Indicator into Respective Sustainability Dimension (Alpha)*

According to the principles of Maqasid al-Shariah, this article posits that the instrument of donating to charity through the Islamic instruments of Sadaqah and Waqf falls under the embellishment category. In Islamic terminology, "Sadaqah" is defined as a voluntary offering of something without expecting a substitute in return with the sole intention of pleasing Allah, whereas "Waqf" refers to a charitable endowment of donating a plot, building, or land. In Islam, charity payment is not mandatory, but it will lead to perfection in society if one pays it. In the context of Islamic banks, paying charity through Sadaqah and Waqf is a part of their operations. Islamic banks are required to channel the income derived from unclear or tainted activities to charitable bodies, including Waqf institutions.

Circulating wealth to the people through charity, Sadaqah and Waqf have upgraded the image of Islamic banks and subsequently improved economic sustainability [19].

- *Alignment of Sustainability Indicator with the UN SDGs (Beta)*

The instrument of charity can also serve multiple UN SDGs. Charity (Sadaqah and Waqf) are used for poverty alleviation and socioeconomic development [47]. In line with that, this study aligned the instrument with UN SDGs 1, 2, 3, and 4. This is because these goals are directly related to poverty alleviation and socioeconomic development.

- *The COVID-19 Response: Based on Alpha and Beta*

According to the new research published by the United Nations University UNU-WIDER (https://www.wider.unu.edu/news/press-release-covid-19-fallout-could-push-half-billion-people-poverty-developing-countries), the pandemic can increase global poverty to about half of a billion people, affecting about 8% of the global population. This increase will affect the UN's agenda of achieving zero poverty by 2030. In addition, according to the world food program, the pandemic may expose around 130 million additional people to hunger by the end of 2020 (https://insight.wfp.org/covid-19-will-almost-double-people-in-acute-hunger-by-end-of-2020-59df0c4a8072). Similarly, it will affect the UN's agenda of achieving zero hunger by 2030. Furthermore, according to the United Nations Educational, Scientific and Cultural Organization (UNESCO) (https://en.unesco.org/covid19/educationresponse), the pandemic has left around 1.6 billion students out of school, which has caused severe mental health issues. Therefore, the instrument of charity by Islamic banks can be targeted towards these goals of no poverty (goal 1), zero hunger (goal 2), good health (goal 3), and quality education (goal 4) to reduce the undesirable impacts of the COVID-19 pandemic. Hence, the following proposition was developed.

**Proposition 5.** *Channeling the amount of charity to UN SDGs 1, 2, 3, and 4 during the pandemic can help poverty alleviation, reduce hunger, promote good health care, and can assist in providing quality education to disadvantaged and deprived stakeholders.*

3.5.6. Disclosure of Earnings Prohibited by Shariah

- *Categorization of Sustainability Indicator into Respective Sustainability Dimension (Alpha)*

According to the principles of Maqasid al-Shariah, this article posits that the disclosure of earnings prohibited by Shariah is complementary to the preservation of faith, self, and wealth. This is because in Islam all financial transactions must be transparent, accurate, and fully recorded [48]. All income received from non-Shariah sources must be fully audited and managed, otherwise they can affect the economic sustainability of Islamic banks [49]. It is the responsibility of the Shariah committee to identify such income and dispatch it to charity funds [50,51].

- *Alignment of Sustainability Indicator with the UN SDGs (Beta)*

The Islamic banking sustainability instrument "the disclosure of earnings prohibited by Shariah" may serve UN SDG 16 and its sub-goal 16.5, which alludes to building transparent, accountable, and effective institutions at all levels. These earnings are channeled by the Islamic banks to charity funds because they violate the principles of Islamic business. Dispatching the earned amount from the profits of Islamic banks ensures greater transparency, accountability, and business effectiveness as well. Consonant with that, this study aligned this instrument with UN SDG 16 and sub-goal 16.5, because that sustainability instrument of Islamic banks will support UN SDG 16 and 16.5.

- *The COVID-19 Response: Based on Alpha and Beta*

Opportunities can be created by investing in policy formulation at the time of crisis. The role of the Shariah committee in dispatching earned money to charity funds based on the violation of business principles is exemplary. It consequently helps build strong

institutions. Strong institutions are vital to fighting the COVID-19 pandemic. Hence, the following proposition was developed.

**Proposition 6.** *Aligning the sustainability instrument "disclosure of earnings prohibited by Shariah" with UN SDG 16 and its sub-target 16.5 will help build transparent, accountable, and effective institutions in fighting against the COVID-19 pandemic.*

*3.6. Environmental Sustainability Indicators*

The categorization, alignment, and policy propositions based on the environmental sustainability indicators are provided below.

3.6.1. Funding for Organizations Upholding a Green Environment

■   *Categorization of Sustainability Indicator into Respective Sustainability Dimension (Alpha)*

The above sustainability indicator is related to the main category of necessities and the subcategories of preservation of posterity and life preservation, based on Maqasid al-Shariah. Jusoff, et al. [52], by quoting from Islamic sources, argued that humans are the stewards of the earth. Matali [53] argued that every Muslim must preserve the ecosystem. Based on the strong existing nexus between the above instrument and green environment it is integrated into the environmental sustainability dimension.

■   *Alignment of Sustainability Indicator with the UN SDGs (Beta)*

This sustainability item may serve multiple UN SDGs related to the environment. This instrument of Islamic banks is linked to supporting the green environment. In line with that, this study proposes aligning this instrument with UN SGDs 7, 13, 14, and 15. This is because these UN SDGs are directly related to preserving the environment. Hence, the alignment will ensure greater compliance by Islamic banks with the sustainable development agenda of the United Nations.

■   *The COVID-19 Response: Based on Alpha and Beta*

The shortage of power can significantly magnify the recovery process from COVID-19. Similarly, the availability of clean water and proper sanitation is vital for maintaining hygiene, which is a fundamental element in fighting the virus. These facts demand Islamic banks to consider climate actions more strictly during their funding process. The time demands that Islamic banks must increase their funding to organizations that focus on renewable energy, clean water, biodiversity, and climate action projects. These fundamental steps will help in curbing the current pandemic of COVID-19 and will limit the chance of future pandemics. Hence, the following proposition was developed.

**Proposition 7.** *Channeling the amount of funding to organizations upholding a green environment to UN SDGs 7, 13, 14, and 15 can lead to the provision of affordable and clean energy, better climate actions, assisting life below water, and more sustainable life on land during the COVID-19 pandemic.*

3.6.2. Amount of Donations to Environmental Awareness

■   *Categorization of Sustainability Indicator into Respective Sustainability Dimension (Alpha)*

This research, based on Maqasid al-Shariah, notes that the above instrument falls within the main category of essential/necessities and the subcategory "preservation of life and preservation of posterity." This is because every Muslim must preserve the ecosystem and planet [53]. Based on the existing nexus between this item and the environment, this study categorized the item into the environmental sustainability dimension.

■   *Alignment of Sustainability Indicator with UN SDGs (Beta)*

This sustainability item may also serve multiple UN SDGs related to the environment. This instrument of Islamic banks is linked to supporting a green environment. In line with that, this study proposes aligning the instrument with UN SGDs 7, 13, 14, and 15. This is because these UN SDGs are directly related to preserving the environment.

Hence, the alignment will ensure greater compliance by Islamic banks with the sustainable development agenda of the United Nations.

■　*The COVID-19 Response: Based on Alpha and Beta*

A healthy ecosystem is our only guarantee to fight the COVID-19 pandemic and our lives depend on the health of the planet. According to the United Nations Educational, Scientific and Cultural Organization (UNESCO) (https://en.unesco.org/news/covid-19 -ocean-ally-against-virus), a bacteria found in the depth of oceans is used for testing to rapidly detect COVID-19. Khalil, et al. [54] argued that higher biodiversity lowers the risk of spreading infectious diseases, a process called the "dilution effect." Therefore, this is the right time for Islamic banks to channel their environmental donations towards the environmental areas identified in the UN SDGs that are sourcing natural material to fight the pandemic. Islamic banks can take the lead in this regard by setting up a special unit devoted to directly channeling environmental donations to the UN SDGs that will ultimately help in fighting the COVID-19 pandemic. Hence, the following proposition was developed.

**Proposition 8.** *Aligning donations to environmental awareness with UN SDGs 7, 13, 14, and 15 can lead to the provision of affordable and clean energy, better climate actions, assist life below water, and promote sustainable life on land during the COVID-19 pandemic.*

3.6.3. Introduction of Green Products and Services

■　*Categorization of Sustainability Indicator into Respective Sustainability Dimension (Alpha)*

Following the principle of Maqasid al-Shariah, the above instrument is related to the main category of necessities and the subcategory of preservation of posterity and preservation of life. This instrument is incorporated into environmental sustainability based on the principle of Maqasid al-Shariah, since Islam commands every Muslim to conserve the ecosystem [53].

■　*Alignment of Sustainability Indicator with the UN SDGs (Beta)*

This instrument of Islamic banks may serve multiple UN SDGs related to the environment. The instrument of green product and services is aimed at supporting the environment. In the same vein, this study posits the alignment between this sustainability instrument with UN SDGs 7, 13, 14, and 15. The proposed alignment will support the sustainable development agenda of Islamic banks based on the UN SDGs related to the environment.

■　*The COVID-19 Response: Based on Alpha and Beta*

The pandemic has allowed an opportunity for every business sector [25] to transform and introduce green products and services in compliance with the UN SDGs. This is because linking products and services with the UN SDGs offers great security, and it is expected that those products and services will be least affected in future pandemics due to their prudence. Sukuk green bonds and green technology financing are efficient Islamic instruments that have the potential to curb the pandemic. Based on the discussion the following proposition was developed.

**Proposition 9.** *Alignment of the introduction of green products and services with UN SDGs 7, 13, 14, and 15 can lead to the provision of affordable and clean energy, climate action, and assist life below water and sustainable life on land during the COVID-19 pandemic.*

*3.7. Social Sustainability Indicators*

The categorization, alignment, and policy propositions based on the social sustainability indicators are provided below.

### 3.7.1. Islamic Training and Education for Staff

▪ *Categorization of Sustainability Indicator into Respective Sustainability Dimension (Alpha)*

Based on Maqasid al-Shariah, the above instrument is complementary to the necessity category of preservation of intellect. This is because the protection of intellect in Islam is not obligatory. However, if Islamic banks operate through education or training to maintain the intelligence of their workers, it would pave the way for them to conduct Islamic banking operations effectively. Julia and Kassim [55] argued that Islamic banks compared to conventional banks are efficiently assisting their staff in the preservation of intellect in terms of awareness of green products. The current study integrated this item into the social sustainability dimension because the impact of banking staff is directly linked to the people and society.

▪ *Alignment of Sustainability Indicator with the UN SDGs (Beta)*

This instrument can target SDG 4 (quality education) and its sub-goal 4.7, which ensures learning new knowledge through collaborative work for sustainable development. As the main aim of the Shariah objectives is to achieve socioeconomic justice and development [28], in line with the Shariah principles, Islamic training and education for staff must ensure quality education aimed at sustainable development. Hence, this alignment will ensure the contribution of the Islamic banks towards SDG 4 and sub-goal 4.7.

▪ *The COVID-19 Response: Based on Alpha and Beta*

The COVID-19 pandemic is unfolding into a great educational crisis. Consonant with that, UNESCO launched global education collation in March 2020 (https://en.unesco.org/news/covid-19-ocean-ally-against-virus) aimed at designing innovative solutions. Islamic banks can avail of this opportunity by encouraging their staff to participate in such collaborative works to learn how the banking staff can play a role in tackling the pandemic at different levels, either directly or indirectly. Through such platforms, Islamic banking staff can also share the Islamic version of the solution to fight the COVID-19 pandemic. This transfer of knowledge and sharing experiences may educate banking staff in sustainably dealing with the virus. Based on the discussion the following proposition was developed.

**Proposition 10.** *Aligning training and education for the staff with UN SDG 4 and sub-goal 4.7 may lead to learning new knowledge through collaborative work that will assist in policy formulation for fighting the COVID-19 pandemic.*

### 3.7.2. Offering Scholarships

▪ *Categorization of Sustainability Indicator into Respective Sustainability Dimension (Alpha)*

According to Maqasid al-Shariah, offering scholarships is considered complementary to the basic category of preservation of intellect. It is incorporated into the social sustainability dimension because providing scholarships to multiple stakeholders will raise the quality of education in society and will uplift social standards as well.

▪ *Alignment of Sustainability Indicator with the UN SDGs (Beta)*

This instrument is directly targeting SDG 4 (quality education) and sub-goal 4B, which focuses on expanding the number of scholarships to developing countries. Therefore, aligning the Islamic banking sustainability instrument of scholarships with UN SDG 4B will promote quality education in supporting the sustainable development agenda of the UN.

▪ *The COVID-19 Response: Based on Alfa and Beta*

According to UNESCO (https://en.unesco.org/covid19/educationresponse), COVID-19 has affected more than one billion students, which accounts for about 67% of total enrolled students globally. There are fears that due to the current turmoil students from mostly underdeveloped or developing countries may completely lose out on their education. This can result in more child-labor cases. Therefore, this is the most appropriate time

for Islamic banks to start offering more scholarships to peoples from mostly underdeveloped countries to avoid child-labor cases. In addition, they should help the world enroll students back to campuses that were affected by the COVID-19 pandemic. Based on the discussion the following proposition was developed.

**Proposition 11.** *Alignment of the sustainability indicator of scholarships with UN SDG 4B during the COVID-19 pandemic will assist students from developing countries in getting a quality education and subsequently mitigate the risk of child labor and child trafficking.*

3.7.3. Approval of New Products and Services by the Shariah Committee

■    *Categorization of Sustainability Indicator into Respective Sustainability Dimension (Alpha)*

Based on the principles of Maqasid al-Shariah, the above sustainability item is linked to the main category of necessities and its subcategory of preservation of faith. It is linked with the preservation of faith because Islam prohibits dealing in haram (prohibited by Islam) products and services. Muslims must also preserve faith while doing business. It is included in the social sustainability dimension because nowadays most of the banking services and products are provided online or through electronic machines that are directly associated with people and society. For instance, the content in the advertisements of Islamic banks must be ethically and socially acceptable to the stakeholders. The delivery of products and services should not lead to any sort of waste generation, carbon footprints, or other forms of social or environmental degradation. Ignoring ethical elements in the delivery of products and services to society may directly affect the social sustainability of Islamic banks. Hence, the Shariah committee must ensure the elements of society before approving any products or services. This is because the intended objectives of the principle of Shariah are to achieve socioeconomic justice for individuals and society and to enhance welfare in society.

■    *Alignment of Sustainability Indicator with the UN SDGs (Beta)*

This instrument may assist in targeting UN SDG 12, which highlights responsible consumption and production. The Shariah committee of Islamic banks in support of Shariah objectives (which aim at achieving socioeconomic justice) are bound to provide approval only for those products and services that achieve socioeconomic justice. Hence, the alignment with UN SDG 12 will promote responsible production and consumption aimed at a sustainable development agenda.

■    *The COVID-19 Response: Based on Alpha and Beta*

The pandemic has allowed an opportunity to shift towards more responsible consumption related to products and services. Humans' wants are unlimited, but the planet has limited resources to fulfil those needs. Humans must appreciate and understand the limits to which they can push nature before it starts to react negatively. Businesses must display those limits in their production and consumption patterns to uplift environment and society. The objectives of Shariah are to achieve socioeconomic justice for individuals and society and to enhance welfare in society. Based on these principles, the Shariah committee prudently monitors any new products and services offered by Islamic banks to promote responsible consumption and production. In a time of crisis, considering the impact of new products and services offered to society and the environment will mitigate the distressing impact of the COVID-19 pandemic in a sustainable manner. Based on the discussion the following proposition was developed.

**Proposition 12.** *The alignment of sustainability indicator "approval by the Shariah committee of new products and services" with UN SDG 12 can reduce the distressing impact of COVID-19 through promoting responsible consumption and production.*

## 4. Conclusions

This article posits an alignment between Islamic banking sustainability indicators and the United Nations Sustainable Development Goals (UN SDGs) to provide timely Islamic-based policy guidelines for reducing the diverse impacts of the COVID-19 pandemic on the triple bottom line (people, planet, and profit) and to achieve sustainable development as well. For this process, in the first place, the article identifies the key Islamic banking sustainability indicators (see Table A2 Appendix A). In the second place, this study categorizes the selected sustainability indicators in the triple bottom line (TBL) in support of the principles of Maqasid al-Shariah and the axial coding method (see Table A3 Appendix A). In the third place, this study establishes the link between the categorized sustainability indicators and the UN SDGs in support of the axial coding method for policy formulation (see Table A4 Appendix A). This study named the new method the ECA method. The new ECA method (see Figure 6) offers a reverse extension to the SDG Compass designed by the Global Reporting Initiative (GRI) to align business policies with the UN SDGs. The SDG Compass is often criticized on the ground that it lacks focus in exploring vital organizational goals, and it only focuses on the implementation phase of the SDGs [39]. Emerging industries such as Islamic banking require detailed prior methodological knowledge such as the exploration and categorization of vital business strategies before moving on to the aligning phase. The new method will assist the Islamic banking industry in a more systematic way of aligning their vital business strategies with the SDGs. The propositions offered should help guide practitioners of the Islamic banking industry to set up a separate sustainability division aimed at integrating Islamic banking sustainability indicators with the UN SDGs with the aim of mitigating the impact of COVID-19 on the triple bottom line. The commentary (in Section 3), can serve as a base in the process of developing the proposed sustainability division in fighting the virus based on Islamic principles. These structural reforms in the short run will assist Islamic banks to play their social role in defeating the virus by offering Islamic solutions, whereas in the long run, they will assist Islamic banks to promote the UN agenda of sustainable development.

### 4.1. Theoretical Implications

In terms of theoretical contribution, this article offers a novel approach for establishing a theoretical connection between sustainability indicators and Maqasid al-Shariah (objectives of Shariah). Secondly, after the ratification of the sustainability indicators, this study linked it with the UN SDGs. Hence, the linkage of Islamic sustainability items with the UN SDGs is dependent on Maqasid-al-Shariah. This highlighted the predominant connection between Maqasid al-Shariah (objectives of Shariah) and the UN SDGs. This study closed the theoretical gap by first identifying and then linking Islamic banking sustainability indicators with the UN SDGs in support of Maqasid-al-Shariah. This made it easier for future researchers to ratify future Islamic banking sustainability indicators accordingly and to link them with the UN SDGs. More importantly, the new ECA method proposed by this study may evolve into an ECA theory for the SDGs in the future with further advancement and refinement.

### 4.2. Social/Practical Implications

In the context of the COVID-19 pandemic, this article offers practical solutions to fight the impact of the crisis faced by the world. The process of aligning Islamic banking sustainability indicators with the UN SDGs will provide a roadmap to recovery from the catastrophe caused by COVID-19. This is because the UN SDG framework is a comprehensive framework covering multifarious aspects for sustainable development. Considering the UN SDGs in terms of the Islamic banking instruments will consequently mitigate the severe impacts of COVID-19 on people–society, planet–environment, and profit–economy. Firstly, it could unlock the potential of the Islamic banking industry (fighting the virus with Islamic instruments) for various stakeholders. Most importantly, it could help the world recover from the distressing impacts of the pandemic in a more sustainable manner.

### 4.3. Policy Recommendations

This study, by proving the anteceding steps for aligning sustainability practices of Islamic banks with the UN SDGs, offers a reverse extension of the SDG Compass designed by the Global Reporting Initiative (GRI) to assist business firms in aligning their business policies with the SDGs. The SDG Compass lacks focus on exploring vital organizational goals, rather only focusing on alignment [39]. Therefore, by proposing the antecedents (step 1 and step 2 in Figure 6), this study offers various policy insights to the practitioners of Islamic banks. In the short run, they must consider the key Islamic banking sustainability indicators (step 1 in Figure 6) in recovery plans. As their alignment is already established with the UN SDGs (steps 2 and 3 in Figure 6), they will mitigate the diverse impacts of the COVID-19 pandemic on the triple bottom line (people, planet, and profit). In the long run, a dedicated SDGs division must be developed in Islamic banks for streamlining (step 1 and step 2 in Figure 6). Achieving the short- and long-run transformation and compliance with the UN SDGs requires an effective governance framework. In the same vein, this study proposes a framework for SDG governance in Figure 7 below.

| **Reinforcing Regulatory System** | **Supervision and Monitoring** | **Educational Activism and Stakeholder Engagement** | **Investing Responsibly in Goods and Services** |
|---|---|---|---|
| • To encourage the adoption of prudent SDG practices by revising the existing regulatory structure of Islamic banks. | • To conduct annual assessments of SDG activities and disclosures to determine the enforcement and consistency of disclosures made and to identify areas for further improvement. | • To conduct programs and commitments for capacity building to enhance SDG culture inside the organization and among the related stakeholders. | • To facilitate investors by providing sustainable goods and services to make responsible investments using effective SDG governance. |

**Figure 7.** Proposed SDG governance framework for the practitioners of Islamic banking for policy.

Figure 7 shows the proposed framework for SDG governance. This will offer policy insights for identification, supervising, and monitoring SDG-related policies at the bank level. Eventually, it will help Islamic banks explore their vital sustainability indicators, categorize them in the triple bottom line, and eventually to align themselves with the UN SDGs to tackle COVID-19 and similar pandemics in the future. Moreover, these transformations will promote sustainable development.

### 4.4. Avenues for Future Research and Recommendations

This article encourages future work on identifying other key Islamic banking sustainability indicators and to establish a link with the UN SDGs using the ECA method. This will assist the practitioners of Islamic banking in their policy formulation. Holistically, it will ensure the achievement of the sustainable development agenda.

**Author Contributions:** Conceptualization, A.J.; methodology, A.J., P.A.A. and R.B.H.; Software, A.J., M.N.M., and J.M.M.; validation, J.M.M., P.A.A. and R.B.H.; formal analysis, A.J.; investigation, A.J. and J.M.M.; resources, J.M.M., M.N.M., data curation, P.N.M.; writing—original draft preparation, A.J.; writing—review and editing, A.J. and P.N.M.; visualization, A.J.; supervision, P.A.A. and R.B.H.; project administration, A.J., J.M.M., and M.N.M.; funding acquisition, J.M.M., M.N.M., and P.N.M. All authors have read and agreed to the published version of the manuscript.

**Funding:** This research received no external funding.

**Conflicts of Interest:** The authors declare no conflict of interest.

## Appendix A

*Appendix A.1. Shortlisting Key Islamic Banking Sustainability Indicators (EXPLORATION)*

To identify the key Islamic banking sustainability indicators, in the first stage, the authors shortlisted the most quoted sustainability measurement indexes that are present in Islamic banking as highlighted in extant literature (refer to Table A1 below). First, the indexes were selected based on the number of citations of scholarly articles and the rankings of the journals. Next, the broader themes of all the indexes were identified. It was found that those indexes mainly used 10 broader themes (see Table A1). In the third step, the dimensions that had frequency distributions above 50 were shortlisted, and further in the process, the top six sustainability dimensions were selected based on their high frequency, i.e., above 50%. Under the six broader themes, the sustainability indicators that appeared in a minimum of three indexes were selected. In the process, the top 12 sustainability indicators were identified (refer to Table A2). The shortlisted sustainability indicators were first verified based on Maqasid-al-Shariah to be in line with Shariah principles and were then separated into the three dimensions of sustainability based on axial coding. Even though the indicators were selected from the indexes used in Islamic banking, their ratification according to Shariah objectives was required. This is because each sustainability index is based on universal (conventional) items and industry-specific items [12]. Therefore, using the principles of Maqasid-al-Shariah, it was assured that the selected sustainability indicators are in line with Shariah objectives and follow the industry-specific Islamic banking philosophy.

Table A1 illustrates the detailed process of segregating the previously used sustainability measurement indexes from Islamic banking into broader themes. It was found that the top sustainability measure indexes have mainly used 10 broader dimensions. Code 1 shows that the index considered sustainability items related to a specific theme and zero otherwise. The transformation process shows that the top sustainability themes for consideration by Islamic banking are "employee," "community and society," "Shariah governance," "products and services," and "environment." The subsequent Table A2 first shows sustainability indicators from the shortlisted broader themes, which involved Islamic principles or Islamic wordings with the indicators. Secondly, it shows the shortlisting of sustainability indicators that appeared in at least three indexes. Details about the shortlisted indicators are presented below in Table A2.

Table A2 shows the selection process of Islamic indicators from the six shortlisted themes based on the frequency distribution. It only shows sustainability indicators from the selected broader themes that involved Shariah principles and were in line with Islamic banking philosophy. Conventional indicators from the previously used indexes were not selected. Only the industry-specific (Islamic banking) indicators were selected. With the help of Maqasid al-Shariah, the shortlisted indicators were ratified to be in line with Shariah principles and subsequently, their linkage with the UN SDGs was established.

*Appendix A.2. Categorizing the Shortlisted Sustainability Indicators (CATEGORIZATION)*

Table A3 shows the detailed categorization process of shortlisted sustainability indicators in the triple bottom line as per the axial coding method.

*Appendix A.3. Alignment of the Categorized Sustainability Indicators with the UN SDGs (ALIGNMENT)*

**Table A1.** Broader themes from previous sustainability indexes used for measuring sustainability practices in the Islamic banking industry.

| Broader Themes from Sustainability Indexes | | Platonova, et al. [22] | Amran, et al. [56] | Aribi and Arun [57] | Mallin, et al. [12] | Aribi and Gao [58] | Farook, et al. [17] | Rahman, et al. [59] | Hassan and Harahap [15] | Othman and Thani [60] | Haniffa and Hudaib [61] | Maali, et al. [62] | Dusuki [63] | Frequency Distribution | Percentage |
|---|---|---|---|---|---|---|---|---|---|---|---|---|---|---|---|
| **1. Employees** | 1. Employment<br>2. Commitment to employees<br>3. Employees<br>4. Employees<br>5. Employees<br>6. Employees<br>7. Employees<br>8. Employees<br>9A. Workers' health and safety<br>9B. Workers' education and training<br>9C. Fair treatment of workers and applicants<br>9D. Fostering Islamic values among staff<br>10. Employees<br>11. Commitment towards employees | 1 | 1 | 1 | 1 | 1 | 1 | 1 | 1 | 1 | 1 | 1 | 1 | 12 | 100 |
| **2. Community and Society** | 1. Commitment to community<br>2. Community development and social goals<br>3. Community<br>4. Society<br>5. Community<br>6. Other aspects of community involvement<br>7A. Financing companies not violating human rights<br>7B. Financing SMEs, providing affordable service to deprived areas<br>7C. Supporting charities and community projects<br>7D. Solving social problems<br>8. Community involvement<br>9. Commitment to society | 1 | 1 | 0 | 1 | 1 | 1 | 1 | 0 | 1 | 1 | 1 | 1 | 10 | 83 |

**Table A1.** *Cont.*

| Broader Themes from Sustainability Indexes | | Platonova, et al. [22] | Amran, et al. [56] | Aribi and Arun [57] | Mallin, et al. [12] | Aribi and Gao [58] | Farook, et al. [17] | Rahman, et al. [59] | Hassan and Harahap [15] | Othman and Thani [60] | Haniffa and Hudaib [61] | Maali, et al. [62] | Dusuki [63] | Frequency Distribution | Percentage |
|---|---|---|---|---|---|---|---|---|---|---|---|---|---|---|---|
| **3. Gover-nance/Shariah Compliance** | 1. Governance 2. Shariah compliance 3. Corporate governance and Shariah-compliant corporate governance 4. Corporate governance 5. BOD and top management 6. Shariah Supervisory Board SSB 7. Sharia opinion —unlawful (haram) transaction 8. Islamic value and SSB 9. Shariah Supervisory Board 10A. Unusual supervisory restrictions 10B. Unlawful (haram) transactions 10C. Sharia Supervisory Council | 0 | 1 | 1 | 1 | 1 | 1 | 1 | 1 | 1 | 1 | 1 | 0 | 10 | 83 |
| **4. Zakat/Charity/ Qard-e-Hasan** | 1. Zakat, charity and benevolent funds 2. Zakat, charity, donations, and Qard-e-Hassan 3. Charity and zakat 4. Zakat, charity, and benevolent loans 5. Zakat, Qard-e-Hassan, Charitable and social activities 6. Paying zakat, charity, and granting Qard-e-Hassan 7. Zakat, charity, and benevolent funds 8. Zakat obligation, Qard fund | 1 | 1 | 1 | 1 | 1 | 1 | 1 | 0 | 0 | 1 | 1 | 1 | 10 | 83 |

**Table A1.** *Cont.*

| Broader Themes from Sustainability Indexes | | Platonova, et al. [22] | Amran, et al. [56] | Aribi and Arun [57] | Mallin, et al. [12] | Aribi and Gao [58] | Farook, et al. [17] | Rahman, et al. [59] | Hassan and Harahap [15] | Othman and Thani [60] | Haniffa and Hudaib [61] | Maali, et al. [62] | Dusuki [63] | Frequency Distribution | Percentage |
|---|---|---|---|---|---|---|---|---|---|---|---|---|---|---|---|
| 5. Product and Services | 1. Products and services 2. Products 3. Products and services 4. Products, services, and fair dealing with supply chain 5. Products and services 6. Products 7. Products and services | 1 | 1 | 0 | 1 | 1 | 0 | 1 | 1 | 1 | 1 | 0 | 0 | 8 | 67 |
| 6. Environment | 1. Environment 2. Environment 3. Environment 4. Environment 5. Environment 6A. Energy and water conservation 6B. Waste recycling policies 6C. Financing companies not harming the environment 7. Environment | 0 | 1 | 1 | 1 | 0 | 0 | 1 | 1 | 1 | 0 | 1 | 1 | 8 | 67 |
| 7. Mission and Vision | 1. Mission and vision statement 2. Strategy—corporate vision 3. Vision and mission statement 4. Vision and mission statement | 1 | 1 | 0 | 1 | 0 | 1 | 0 | 0 | 0 | 1 | 0 | 0 | 5 | 42 |
| 8. Customer and Clients | 1. Ethical behavior, stakeholders' engagement, and customer relations 2. Listening to public view and concern, fostering Islamic values among customers 3. Customers 4. Late repayments and insolvent clients | 0 | 0 | 0 | | 1 | | 0 | 1 | 1 | 0 | 0 | 1 | 4 | 33 |
| 9. Debtors | 1. Commitment to debtors 2. Debtors 3. Debtors 4. Commitment to debtors | 1 | 0 | 1 | 1 | 0 | 1 | 0 | 0 | 0 | 1 | 0 | 0 | 5 | 42 |
| 10. Other | 1. Finance and Investment 2. Contribution | 0 | 0 | 0 | | 0 | | | 0 | 1 | 0 | 0 | 0 | 1 | 8 |

**Table A2.** Shortlisted Islamic indicators from broader sustainability themes used in the Islamic banking industry.

| Broader Themes | Platonova, et al. [22] | Amran, et al. [56] | Aribi and Arun [57] | Mallin, et al. [12] | Aribi and Gao [58] | Farook, et al. [17] | Rahman, et al. [59] | Hassan and Harahap [15] | Othman and Thani [60] | Haniffa and Hudaib [61] | Maali, et al. [62] | Shortlisted Sustainability Indicators |
|---|---|---|---|---|---|---|---|---|---|---|---|---|
| **1. Employees** | Training: Shariah awareness | Training: Shariah awareness | The policy on education and training in relation to the Islamic financial institution | Employee training and development in line with Islamic principles | 0 | 0 | Training: Shariah awareness | 1A. Religious freedom for Muslims to perform prayers. 1B. The proper place of worship for employees. | Training: Shariah awareness | The policy on education and training of employees in line with Islamic principles | Shariah education for the employee | 1. Islamic training and education for the staff (8 indexes). |
| **2. Community and Society** | Conferences on Islamic economics and other educational areas | Zakat. Qard-e-Hassan and Sadaqah -for strategic social development. | 0 | Zakat | 0 | 0 | 0 | Scholarships. Sadaqah/Waqf/Qard-e-Hassan. | Conferences on Islamic economics | 0 | Supporting charities and community projects | 2. Sadaqah, charity, Qard-e-Hassan (4 indexes). 3. Offering scholarships, conducting Islamic conferences (3 indexes). |
| **3. Governance/ Shariah Compliance** | 0 | 1A. Nature of unlawful transactions. 1B. Allocation of profits based on Shariah principles. 1C. Shariah screening of investments. 1D. Zakat. calculation and payment | 1A. Nature of unlawful transactions. 1B. Compliance with Shariah in all products and services. | Commitment to ethical conduct | 1A. Report of SSB. 1B. Nature of unlawful transactions or services. | 1A. Unlawful haram transactions. 1B. Shariah supervisory council. 1C. Unusual supervisory restrictions. | 1A. Nature of unlawful transactions. 1B. Allocation of profits based on Shariah principles. 1C. Shariah screening of investments. 1D. Zakat calculation and payment | Declaration of forbidden activities | SSB Report | Nature of unlawful transactions | Fostering Islamic values among staff | 4. Disclosure of earnings prohibited by Shariah (7 indexes). 5. Shariah screening of investments (3 indexes). 6. Allocation of profits based on Shariah principles (3 indexes). |
| **4. Zakat/Charity/ Qard-e-Hassan** | Zakat, charity, and benevolent funds (Qard-e-Hassan) | Zakat, charity, and benevolent funds (Qard-e-Hassan) | Zakat, charity, and benevolent funds (Qard-e-Hassan) | Charity and zakat | Zakat, charity, and benevolent funds (Qard-e-Hassan) | Zakat, and (Qard-e-Hassan) | 0 | 0 | Zakat, charity, and benevolent funds (Qard-e-Hassan) | Zakat, charity, and benevolent funds (Qard-e-Hassan) | Zakat, charity, and benevolent funds (Qard-e-Hassan) | 7. Zakat payment (9 indexes). 8. Charity/Sadaqah (8 indexes). 9. Qard-e-Hassan (benevolent funds) (8 indexes). |

**Table A2.** *Cont.*

| Broader Themes | Platonova, et al. [22] | Amran, et al. [56] | Aribi and Arun [57] | Mallin, et al. [12] | Aribi and Gao [58] | Farook, et al. [17] | Rahman, et al. [59] | Hassan and Harahap [15] | Othman and Thani [60] | Haniffa and Hudaib [61] | Maali, et al. [62] | Shortlisted Sustainability Indicators |
|---|---|---|---|---|---|---|---|---|---|---|---|---|---|
| **5. Products and Services** | 1A. No involvement in non-permissible activities. 1B. Approval ex-ante by SSB for new product. | 1A. Introduction of SSB-approved new product. 1B. Basis of Shariah concept on new products. | 0 | New product and services in maintenance with religious credentials | 0 | Products and services in line with Shariah principles | 1A. Introduction of SSB-approved new product. 1B. Basis of Shariah concept on new products. | Halal status of the product | 1A. Approval ex-ante by SSB for the new product. 1B. Basis of Shariah concept on new products | 0 | 0 | 10. Approval of new products and services by the Shariah Committee (7 Indexes). |
| **6. Environment** | 0 | 1. Introduction of green product. 2. Amount of donations to environmental awareness. | Lending policy | 0 | 0 | 0 | 1. Introduction of green product. 2. Amount of donations to environmental awareness. 3. Investment in sustainable development projects. | Environmental education | 0 | 1A. The amount and nature of any donations or activities undertaken to protect the environment. 1B. The projects financed by the bank that may lead to harming the environment. | Financing companies not harming the environment | 11. Funding for organizations upholding a green environment (4 indexes). 12. Amount of donations to environmental awareness (3 indexes). 13. Introduction of green products and services (3 indexes). |

**Table A3.** Four-step axial coding for categorizing sustainability indicators into three dimensions of sustainability.

| Items | 1: Phenomena | 2: Causal Condition | 3: Intervening Strategies | 4: Consequences | Category |
|---|---|---|---|---|---|
| | *Q1: How are the item and category related to each other?* | *Q2: How do the item and category influence each other?* | *Q3: What are the actions and strategies required to relate item and category?* | *Q4: What are the consequences of relating item and category?* | |
| 1. Shariah screening of investments | The item and category are related to each other through their economic kind of nature. The investments are made through the funds available to the Islamic banks in the form of different economic capitals. The investment of Islamic banks is recorded in the annual reports of the Islamic banks, which depicts the economic position of banks. Hence, based on the economic nature this item and category are closely related to each other. | The item and category greatly affect each other based on inductive information. This is because it will increase the trust of stakeholders in Islamic banks to be more in line with Islamic values by executing and reporting on this item. The increased trust of stakeholders would eventually lead the Islamic bank to raise more funds. This would gradually improve its economic viability. Thus the object and category favourably affect each other based on the positive causal situation. | The strategy adopted by this study to relate the item to economic sustainability is to establish its theoretical relation. Following the Maqasid al-Shariah principle, this study ponders that the investment screening of Shariah falls under the "essential" category and "preservation of faith and wealth" subcategory. Based on inductive knowledge and Islamic teachings, preservation of wealth according to Islamic teachings will improve economic sustainability. Hence, as a strategy and action, Islamic banks are required to intensify the level of Shariah screening of investments for better economic sustainability. | Based on inductive knowledge, relating item and category with each other will improve the economic sustainability profile of the Islamic banks. This will offer Islamic banks greater economic surveillance and better management of Islamic funds in compliance with Shariah principles. | **Economic Sustainability Dimension** |
| 2. Allocation of profits based on Shariah principles | This item and category relate to each other based on their economic (monetary) nature. Profits are paid from the earnings of banks, recorded in the income statement. The income statement shows the economic condition of the banks. Hence, this item and category follow the same philosophy and are related to each other in monetary terms. | As per Islamic law, Islamic banks are expected to allocate profit to all depositors with complete fairness, and also to protect their capital in the process. The dedication of such actions would increase the customer's interest in the Islamic bank, which will increase deposits. It would thus boost the economic viability of Islamic banks. Thus the item and category favorably affect each other based on the causal situation. | This is based on the strategy of establishing a theoretical link between the item and category. The principles of Maqasid al-Shariah alludes to the allocation of profits based on Shariah principles falling under the "essential" category and "preservation of wealth" subcategory. Based on inductive knowledge, better wealth preservation will improve economic sustainability accordingly. Hence, the action and strategies required by Islamic banks are to further ensure the allocation of effective Shariah principles to safeguard economic sustainability. | As a consequence of relating this item with the category, it will safeguard the economic sustainability of Islamic banks. As a causal condition, following Shariah principles in profit sharing will positively address the stakeholders, which may increase the cash inflow as a positive goodwill of the banks. As a consequence, economic sustainability will be improved. | |

**Table A3.** *Cont.*

| | | | | |
|---|---|---|---|---|
| 3. Qard-e-Hassan | This item is related to the economic sustainability dimension because of its monetary nature. This is because Qard-e-Hassan is paid as an interest-free loan from the economic profit of banks. The bank with the most economic funding (profits) can distribute more Qard-e-Hassan and vice versa. Therefore, this item and category are related to each other in terms of the same segment of recording, i.e., financial statements. | The Qard-e-Hassan financing facility is more applicable in poor or underdeveloped countries where it can serve to remove hardship from society and life. Providing Qard-e-Hassan to poor customers will increase the goodwill of the Islamic banks not only in their customers' minds but also in public at large. As a result, Islamic banks will attract more deposits from other stakeholders, which may strengthen their economic sustainability positively. Hence, based on the causal condition the item and category positively influence each other. | According to the principles of Maqasid al-Shariah, this study categorized Qard-e-Hassan as a complementary item to the subcategories of preservation of self or life and preservation of posterity. This is based on inductive knowledge when the self, life, and posterity of people are preserved. The economic burden on them is reduced. Hence, to improve the economic sustainability of Islamic banks' stakeholders and the banks themselves, these principles can serve as a strategic base. | Linking Qard-e-Hassan with economic sustainability will improve economic sustainability. One would argue that paying interest-free loans should decrease economic sustainability in the short run. However, holistic inductive knowledge would argue that it will improve economic sustainability in the long run. This is because of the goodwill philosophy. |
| 4. Charity—Sadaqah—Waqf | The items of charity and economic sustainability are closely related to each other based on their monetary nature. Charity is paid from the different banking sources and funds, which comes under the economic head of the annual reports. Therefore, this item and category both share the same financial head under the annual report of the Islamic banks and are strongly related to each other. | In the context of Islamic banks, paying charity through Sadaqah and Waqf is a part of their operations. Islamic banks are required to channel the income derived from unclear or tainted activities to charitable bodies, including Waqf institutions. Circulating wealth to the people through charity, Sadaqah and Waqf has upgraded the image of Islamic banks and subsequently, improved their economic sustainability. Hence, based on the causal condition the item and category positively influence each other. | According to the principles of Maqasid al-Shariah, this study posits that the instrument of donating to charity through the Islamic instruments of Sadaqah and Waqf falls under the "embellishment" category. Paying charity through Sadaqah and Waqf is a part of their operations. Islamic banks are required to channel the income derived from unclear or tainted activities to charitable bodies, including Waqf institutions. Circulating wealth to the people through charity, Sadaqah, and Waqf has upgraded the image of Islamic banks and subsequently improved economic sustainability. Hence, as an efficient strategy and action, Islamic banks must channel charity to the least addressed SDGs to promote sustainable development, which as a consequent will improve their economic sustainability based on the goodwill and compliance philosophy. | The linkage of charity with the economic sustainability dimension will improve economic sustainability. In line with the philosophy of Qard-e-Hassan, in the short term charity payment decreases the profit of Islamic banks, which can be perceived as a negative impact on economic sustainability. However, in the long run, based on the goodwill philosophy, the instrument of charity payment will improve the economic sustainability of Islamic banks as a consequence of receiving more funds through their positive goodwill. |

**Economic Sustainability Dimension**

**Table A3.** *Cont.*

| | | | | |
|---|---|---|---|---|
| 5. Disclosure of earnings prohibited by Shariah | Based on Islamic philosophy the earning of income by prohibited sources is not allowed. Therefore, if any bank is willingly dealing in it and is not recording it, it can affect the economic condition of an Islamic bank negatively and vice versa. Hence, the progression of the economic performance of Islamic banks lies in avoiding haram profit. If it is committed mistakenly, the bank must immediately send it to charity. Hence, this item is strongly related to the economic sustainability dimension based on the Shariah philosophy. | To prevent the recurrence of such a transaction, Islamic banks must design control systems and forward any other gain to the funds of charities. This will strengthen the client's confidence in Islamic banks. More deposits will be produced with improved confidence, and more deposits will boost the economic viability of Islamic banks. Therefore the item and category positively affect each other based on the positive causal situation. | As a strategy, to develop the theoretical link between the item and category, based on the principles of Maqasid al-Shariah, this study posits that the disclosures of earnings prohibited by Shariah are complementary to the preservation of faith, self, and wealth. This is because in Islam all financial transactions must be transparent, accurate, and fully recorded. All income received from non-Shariah sources must be fully audited and managed, otherwise they would affect the economic sustainability of the Islamic bank. Hence, as a strategy and action, Islamic banks must properly channel these earnings to charity funds to preserve their economic sustainability. | The consequences of relating this item with economic sustainability may be seen in the shape of strong economic sustainability. This is because the recording of unlawful and haram income and simultaneously dispatching it to a charity fund will increase the Shariah rating process of Islamic banks, which as a consequent will keep the existing economic stakeholders satisfied, and will attract more customers, which will increase the economic sustainability of Islamic banks. |
| 6. Zakat payment | Using inductive knowledge, zakat is related to the economic category based on the fact that zakat is paid from the income earned and is reported on the economic part of the annual report, i.e., on the income statement. Hence, both are related to each other based on their financial nature. | Based on inductive knowledge, zakat item zakat and the economic category strongly influence each other, i.e., zakat payment increases the goodwill of Islamic banks. As an effect of high goodwill, the banks generate more funds, which eventually influence their economic sustainability positively. | The strategy adopted by this study to relate the item with the category is by developing a theoretical link. According to Maqasid al-Shariah, zakat falls under the subcategory of the preservation of wealth. As a strategic requirement, the principles assure a link in the presence of Shariah principles. Now as an action, the bank must channel its zakat payment to SDGs that are relatively unaddressed. As a result of such actions and compliance, the economic sustainability of Islamic banks will get better. | The consequences of relating zakat with the economic category are assumed to be positive. Based on the goodwill philosophy (causal condition), it improves the economic sustainability of Islamic banks. Strong economic sustainability ultimately improves the financial performance of Islamic banks in a positive way. Hence, consequently, the relative results are positive. |

**Economic Sustainability Dimension**

Table A3. *Cont.*

| | | | | |
|---|---|---|---|---|
| 7. Quota funding to organizations not harming a green environment | Islam considers humans the stewards of the earth. Based on this, every business that operates under Islamic principles must preserve the ecosystem. Therefore, based on the Shariah philosophy of stewardship, this funding is related to environmental sustainability. | The causal condition between this funding and environmental sustainability is significant. The low funding for organizations that are involved in renewable energy projects will lead to a lower level of green projects, which as a result will reduce environmental sustainability and vice versa. Hence, the causal condition between this item and category depends on the level of funds. If there are more funds, environmental sustainability will be better and vice versa. | Based on the theory of Maqasid al-Shariah, the above instrument is linked to the main category of "essential" and subcategory of preservation of posterity and preservation of life. Hence, it is assumed that there is a theoretical link between the item and category in the context of Shariah. The action required by Islamic banks is to prioritize funding for better environmental preservation. | The consequence of relating this funding with environmental sustainability will protect the environment. This is because that funding as a consequence will accelerate green projects, which will positively protect the environment. |
| 8. Amount of donations given to environmental awareness | This item also relates to the environmental sustainability dimension based on the Islamic philosophy of humans as stewards of the earth. This philosophy triggers the banks to distribute donations to environmental awareness. Hence, this item and category are related to each other based on the Islamic concept of stewardship. | The causal condition between the funding given for environmental awareness and the environmental sustainability dimension depends on the level of funds. If the amount of funds is high, the causal condition would be high. If the amount is less the causal condition would be negative. | According to the principles of Maqasid al-Shariah, this item is categorized as the preservation of self/life and the preservation of posterity under the main category of "essential" because there exists a theoretical link and consensus between this item and the principles of Shariah for improving environmental sustainability. Therefore, as a strategy and action, Islamic banks are required to increase the amount of such donations. | The consequence of relating these donations to environmental sustainability will protect the environment and will improve environmental sustainability ratings if the funding amount is high and vice versa. **Environmental Sustainability Dimension** |
| 9. Introduction of green products and services | The item and the category of environmental sustainability relate to each other based on the nature of service and product initiation. This is because the purpose of launching the product or service is to keep the environment in nature. Hence, based on the purpose, this item and category relate to each other. | The causal condition between this item and category is significant in terms of impact. The offered product and service would create a positive impact on the environment based on its purpose of serving the environment. Hence the causal condition is perceived as positive. | According to the theory of Maqasid al-Shariah, this item is categorized as the preservation of self/life and the preservation of posterity under the main category of "essential" because there exists a theoretical link and consensus between this item and the principles of Shariah for improving environmental sustainability. Therefore, as a strategy and action, Islamic banks are required to increase the production and services of green products. | The consequence of relating this item to environmental sustainability will bring positive environmental ratings. This is because those products and services as a consequence will accelerate green projects, which will protect the environment. |

**Table A3.** *Cont.*

| | | | | |
|---|---|---|---|---|
| 10. Islamic training and education for staff | This item is related to the social sustainability dimension based on the nature of the work. That is, Islamic training and education for staff would ensure decent work practices ensured by Shariah. As Islam prohibits discrimination based on gender, race, or ethnicity, the concept is referred to as Husn-e-akhlaq/obligingness. | The causal condition between this item and category depends upon the standard of training and education. If the standard of Islamic education is in-depth and pure, it will boost the character of banking staff positively towards their co-workers, which as a cause will affect social sustainability positively. | Islamic training and education are categorized as a complimentary item to the preservation of intellect under the category of "essential." Based on the consensus and theoretical link between these items, Islamic banks as a strategy are required to set up a banking institution dedicated to supporting social justice and social up-gradation through banking staff. | The linking of Islamic training and education for staff with social sustainability as a consequence will bring positive outcomes in the social dimension of sustainability. | |
| 11. Scholarships | This is related to the social sustainability dimension because providing scholarships to multiple stakeholders will raise the quality of education in society and will uplift social standards as well. | The causal condition of this item with that of social sustainability is also perceived as positive. This is because providing scholarships will uplift the standard of education of different stakeholders, and those stakeholders using their knowledge and education may work for the betterment of society based on the social contract theory. | According to the theory of Maqasid al-Shariah, this item of scholarships is categorized as a complimentary item to the preservation of intellect under the main category of "essential" because there exists a theoretical link between this item and Shariah principles. So, as a way forward, Islamic banks must increase the amount of funding for scholarships to reduce child labor and forced labor. Eventually, it will improve the social sustainability of Islamic banks. | There are fears that due to the current turmoil of COVID-19, students from mostly underdeveloped or developing countries may completely lose out on their education. This can result in more child-labor cases. Therefore, this is the most appropriate time for Islamic banks to start offering more scholarships to peoples from mostly underdeveloped countries to avoid child-labor cases. Hence, the consequences of offering scholarships on social sustainability would be very positive. | **Social Sustainability Dimension** |

**Table A3.** *Cont.*

| 12. Product and service labeling (approved by the Shariah Committee) | This is related to social sustainability because Islam prohibits dealing in haram products and services. Muslims must preserve faith while doing business. Hence, based on the principles of Islamic faith, this item and category are related to each other. | The causal condition between this item and social sustainability is also perceived positively. This is because unethical labeling that can affect any people from any religion may cause high distress among the stakeholders of Islamic banks. Therefore, the causal condition between this item and category is significant. | According to the theory of Maqasid al-Shariah, the product and service approval from the Shariah Committee is categorized as essential under the subcategory of preservation of faith because there exists a theoretical link between this item and Shariah principles. So, as a way forward, as a strategy and action, Islamic banks are required to deepen the process of Shariah evaluation to safeguard its social sustainability. | Nowadays most banking services and products are provided online or using electronic machines that are directly associated with people and society. For instance, the content in the advertisements of Islamic banks must be ethically and socially acceptable for all stakeholders from all religions. Ignoring ethical elements in the delivery of products and services to society may directly affect the social sustainability of Islamic banks. Hence, the Shariah Committee must ensure the elements of society before approving any product or service. This is because the intended objectives of the principle of Shariah are to achieve socioeconomic justice for individuals and society and to enhance welfare in society. Hence, the consequence of this linkage is positive. | **Social Sustainability Dimension** |

**Table A4.** Four steps of axial coding for aligning sustainability indicators with the UN SDGs.

| Items | 1: Phenomena | 2: Causal Condition | 3: Intervening Strategies | 4: Consequences | Alignment with the UN SDGs |
|---|---|---|---|---|---|
| | *Q1: How are the item and category related to each other?* | *Q2: How do the item and category influence each other?* | *Q3: What are the actions and strategies required to relate item and category?* | *Q4: What are the consequences of relating item and category?* | |
| 1. Shariah screening of investments | The item and UN SDG are related to each other based on the principles of transparency. Shariah screening of investments offers great transparency and accountability based on Islamic laws in developing effective institutions. Shariah objectives aim to promote social welfare (Al-Maslahah), therefore, the instrument of Shariah screening of investments will ensure the prevention of investments in inappropriate haram business (forbidden by Islamic laws) such as gambling, which generally violates the business objectives of free and fair exchange, which halts the process of building strong and transparent institutions. | The causal condition between the sustainability indicators and UN SDG 6 and sub-goal 16.6 is considered positive. This is because Shariah objectives aim at achieving socio-economic development. | As a strategic requirement, this study proposes a modern governance role, i.e., SDG governance (refer to Figure 7). The modern governance role ensures the alignment process between sustainability indicators and the UN SDGs in four detailed stages. | As a consequence of relating the item and category, the compliance of Islamic banks with the SDG will increase, which will promote sustainable development and will mitigate the stress from COVID-19 on the triple bottom line. | SDG 16 (sub-goal 16.6) |
| 2. Allocation of profits based on Shariah principles | The item and UN SDG are related to each other in terms of principles and laws, as UN SDG 10.3 alludes to reducing inequality of outcomes by eradicating discriminatory principles and laws. On the other hand, Shariah principles (based on the principles of socio-economic justice) reduce inequalities of outcomes by fairly distributing profit to all stakeholders. | The causal condition between the item and SDG is considered positive. This is because the allocation of profits based on Shariah principles will enhance compliance of Islamic banks with SDG goal 10 and its sub-goal 10.3. | As a strategic requirement, this study proposes a modern governance role, i.e., SDG governance (refer to Figure 7). The modern governance role ensures the alignment process between sustainability indicators and the UN SDGs in four detailed stages. | As a consequence of relating the item and category, the compliance of Islamic banks with the SDG will increase, which will promote sustainable development and will mitigate the stress from COVID-19 on the triple bottom line. | SDG 10 (sub-goal 10.3) |

<div align="center">**Table A4.** *Cont.*</div>

| | | | | |
|---|---|---|---|---|
| 3. Qard-e-Hassan | The item and UN SDGs are related to each other in terms of their philanthropic nature and addressing the needs of poor stakeholders. This study relates the instrument with multiple SDGs because these goals are interconnected. Improvement in one goal, for instance no poverty, as a subset, will bring an improvement in the other goals as well, such as alleviating poverty enhancing the purchasing power of stakeholders, which allows them to afford food, health services, water, and energy. | The causal condition between the item and multiple SDGs is perceived as positive. This is because the item will address the needs of various deprived stakeholders related to SDGs 1, 2, 3, 4, 6, 7, 10, and 15, which as a result will reduce the challenges faced by those deprived stakeholders. | As a strategic requirement, this study proposes a modern governance role, i.e., SDG governance (refer to Figure 7). The modern governance role ensures the alignment process between sustainability indicators and the UN SDGs in four detailed stages. | As a consequence of relating the item and category, the compliance of Islamic banks with the SDGs will increase, which will promote sustainable development and will mitigate the stress from COVID-19 on the triple bottom line. | SDGs 1, 2, 3, 4, 6, 7, 10, 15 |
| 4. Charity—Sadaqah—Waqf | The item and UN SDGs are related to each other in terms of their philanthropic nature and addressing the needs of poor stakeholders. Charity (Sadaqah and Waqf) are used for poverty alleviation and socioeconomic development. In line with that, this study related the instrument with UN SDGs 1, 2, 3, and 4. This is because these goals are directly related to poverty alleviation and socioeconomic development. | The causal condition between the item and UN SDGs is perceived as positive. This is because channeling charity towards the goals will address the needs of the deprived stakeholders. | As a strategic requirement, this study proposes a modern governance role, i.e., SDG governance (refer to Figure 7). The modern governance role ensures the alignment process between sustainability indicators and the UN SDGs in four detailed stages. | As a consequence of relating the item and category, the compliance of Islamic banks with the SDGs will increase, which will promote sustainable development and will mitigate the stress fromCOVID-19 on the triple bottom line. | SDGs 1, 2, 3, 4 |
| 5. Disclosure of earnings prohibited by Shariah | The item and UN SDG are related to each other based on the principles of transparency. UN SDG 16.5 alludes to building transparent, accountable, and effective institutions at all levels. The earnings are channeled by Islamic banks to charity funds because they violates the principles of Islamic business. Dispatching the earned amount from the profit of Islamic banks ensures greater transparency, accountability, and business effectiveness as well. Consonant with that, this study relates the instrument to UN SDG 16 and 16.5. | The causal condition between the item and UN SDG is also significantly positive. This is because disclosures of earnings prohibited by Shariah will ensure business transparency, which helps build a strong institution. | As a strategic requirement, this study proposes a modern governance role, i.e., SDG governance (refer to Figure 7). The modern governance role ensures the alignment process between sustainability indicators and the UN SDGs in four detailed stages. | As a consequence of relating the item and category, the compliance of Islamic banks with the SDG will increase, which will promote sustainable development and will mitigate the stress from COVID-19 on the triple bottom line. | SDG 16 (Sub-goal: 16.6) |

Table A4. *Cont.*

| | | | | |
|---|---|---|---|---|
| 6. Zakat payment | The item and UN SDGs are related to each other in terms of their philanthropic nature and addressing the needs of poor stakeholders. Zakat is paid with the aim of supporting disadvantaged people in society (Malik, 2016). In the same vein, this study relates the instrument of zakat to the UN SDGs because these goals are directly related to the needs of disadvantaged people. | The causal condition between the item and UN SDGs is also perceived as positive. This is because channeling zakat towards the goals will address the needs of the deprived stakeholders. | As a strategic requirement, this study proposes a modern governance role, i.e., SDG governance (refer to Figure 7). The modern governance role ensures the alignment process between sustainability indicators and the UN SDGs in four detailed stages. | As a consequence of relating the item and category, the compliance of Islamic banks with the SDGs will increase, which will promote sustainable development and will mitigate the stress from COVID-19 on the triple bottom line. | SDGs 1,2,3,4,6,7,10,15 |
| 7. Funding organizations not harming a green environment | This instrument and the UN SDGs are related to each other based on the principles of protecting the environment. | This instrument and the UN SDGs positively affect each other. Increase in this instrument will bring positive outcomes to the SDGs. Hence, the causal condition is perceived as positive. | As a strategic requirement, this study proposes a modern governance role, i.e., SDG governance (refer to Figure 7). The modern governance role ensures the alignment process between sustainability indicators and the UN SDGs in four detailed stages. | As a consequence of relating the item and category, the compliance of Islamic banks with the SDGs will increase, which will promote sustainable development and will mitigate the stress from COVID-19 on the triple bottom line. | SDGs 7,13,14,15 |
| 8. Amount of donations given for environmental awareness | This instrument and the UN SDGs are related to each other based on the principles of protecting the environment. | This instrument and the UN SDGs positively affect each other. Increase in this instrument will bring positive outcomes to the SDGs. Hence, the causal condition is perceived as positive. | As a strategic requirement, this study proposes a modern governance role, i.e., SDG governance (refer to Figure 7). The modern governance role ensures the alignment process between sustainability indicators and the UN SDGs in four detailed stages. | As a consequence of relating the item and category, the compliance of Islamic banks with the SDGs will increase, which will promote sustainable development and will mitigate the stress from COVID-19 on the triple bottom line. | SDGs 7,13,14,15 |

**Table A4.** *Cont.*

| | | | | |
|---|---|---|---|---|
| 9. Introduction of green products and services | This instrument and the UN SDGs are related to each other based on the principles of protecting the environment. | This instrument and the UN SDGs positively affect each other. Increase in this instrument will bring positive outcomes to the SDGs. Hence, the causal condition is perceived as positive. | As a strategic requirement, this study proposes a modern governance role, i.e., SDG governance (refer to Figure 7). The modern governance role ensures the alignment process between sustainability indicators and the UN SDGs in four detailed stages. | As a consequence of relating the item and category, the compliance of Islamic banks with the SDGs will increase, which will promote sustainable development and will mitigate the stress from COVID-19 on the triple bottom line. | SDGs 7,13,14,15 |
| 10. Islamic training and education for staff | This instrument and UN SDG are related to each other in terms of the category of education. | The causal condition between the instrument and the UN SDG is perceived as positive. The training will serve UN SDG 4 and 4.7 positively. | As a strategic requirement, this study proposes a modern governance role, i.e., SDG governance (refer to Figure 7). The modern governance role ensures the alignment process between sustainability indicators and the UN SDGs in four detailed stages. | As a consequence of relating the item and category, the compliance of Islamic banks with the SDG will increase, which will promote sustainable development and will mitigate the stress from COVID-19 on the triple bottom line. | SDG 4 (Sub-goal: 4.7) |
| 11. Scholarships | This instrument and UN SDG are related to each other in terms of the category of education | The causal condition between the instrument and the UN SDG is perceived as positive. The training will serve UN SDG 4 and 4.7 positively. | As a strategic requirement, this study proposes a modern governance role, i.e., SDG governance (refer to Figure 7). The modern governance role ensures the alignment process between sustainability indicators and the UN SDGs in four detailed stages. | As a consequence of relating the item and category, the compliance of Islamic banks with the SDG will increase, which will promote sustainable development and will mitigate the stress from COVID-19 on the triple bottom line. | SDG 4 (Sub-goal: 4B) |
| 12. Product and service labeling (approved by the Shariah Committee) | This instrument and category are related to each other in terms of product and service responsibility aimed at society. As such Shariah approval will promote responsible production and consumption. | The causal condition between the item and UN SDG is positive. Shariah approval will promote responsible consumption and production. Hence, it will support UN SDG 12 positively. | As a strategic requirement, this study proposes a modern governance role, i.e., SDG governance (refer to Figure 7). The modern governance role ensures the alignment process between sustainability indicators and the UN SDGs in four detailed stages. | As a consequence of relating the item and category, the compliance of Islamic banks with the SDG will increase, which will promote sustainable development and will mitigate the stress from COVID-19 on the triple bottom line. | SDG 12 |

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
