# Peer review of "Alignment of Islamic Banking Sustainability Indicators with Sustainable Development Goals: Policy Recommendations for Addressing the COVID-19 Pandemic"

_sustainability, doi:10.3390/su13052607_

Round 1
Reviewer 1 Report
Although I believe that this paper deals with an important topic and may contribute to policy development, I have some concerns about it.
Particularly, authors have focused their paper on the presentation of results rather than the methodology they implemented to align Islamic Banking Indicators with UN Sustainable Development Indicators. This methodology remains very cryptic to most of readers who are not familiar with these issues. In the paper only Figure 4 shortly presents the methodology, while the Appendix shows results. My suggestion is to expand section 3. Please, consider the following issues: who was involved in the alignment process? Are there any other methodologies that can lead to the same or similar results? Why do authors believe that results are acceptable?
Please, include a sound discussion of policy implications in the last section.
See lines 164-165: “The theoretical foundation for establishing the link between the shortlisted sustainability indicators (Refer Appendix Table B) with the UN-SDGs as shown in Figure 2 above.” This sentence is unclear.
Author Response
Dear Respected Reviewer 1:
I really appreciate your insightful comments and suggestions. All of the said changes have been addressed accordingly. Kindly refer to the attached (rebuttal/corrections file). Thank you!
Kind Regards:

Reviewer 2 Report
Dear Authors,
The triple bottom line approach to sustainability is a scientifically highly questionable concept. I suggest an analysis based on more complex concepts.
e.g.https://www.mdpi.com/2079-9276/8/4/159
I strongly recommend the application of the 'doughnut economics' concept: https://www.kateraworth.com/doughnut/
It is also important to know that each sustainability goal is not independent of each other. This hierarchy should be used in the analysis: https://www.mdpi.com/2079-9276/8/2/101
In my opinion, the format of the literature references is not appropriate.
266: 'Error! Reference source not found' please correct it.
Author Response
Dear Respected Reviewer 2:
I really appreciate your insightful comments and suggestions. All of the said changes have been addressed accordingly. Kindly refer to the attached (rebuttal/corrections file). Thank you!
Kind Regards:

Reviewer 3 Report
It is not clear by the date on which the data is given in Table 1 and Table 2. It should be indicated both in the title of tables and in text. If the report was published in 2019 it does not mean that data concerns 2019 but 2018 year.
It should be rethinked whether the basic principles of SDGs (Figure 2) are required in text.
Authors should refer to the Glaser and Strauss, who first described the Grounded Theory in 1967 (point 3.1.). The most valuable is Figure 6 and Tables A, B,,C, D.
The term “ECA theory” is overused. It should be indicated that whis is the implementation of the ECA method for receiving sustainable development goals, since it is not a theory (especially this is not a “new” theory). Authors often confused the concept of theory and method.
in point 4.1.1. Authors should justified by numbers what are the positiv affects of economic sustainability of the Islam banks.
In point 4.1.2. it should be justified by showing the statistical data how Islamic banks reduce inequalities during the pandemic (the same is crucial in other parts of the paper with similar statements or hypotheses). What has changed in the isalmic banking sector after the beginning of pandemic (it is worth indicating the financial indexes like profitability, efficiency, volume of loans (i.g. Qard-e-Hasan, Sadaqah, Waqaf) in Muslim countries, changes in volume of Zakat payment, the change of number of customers etc. ?
The reviewer is not convinced that Maqasid Al-Shariah is a theory - it is rather philosophy than theory. There is a lack of direct impacts of the COVID-19 pandemic on the triple bottom line (people, planet and profit) supported by statistical data.
Furthermore, the language requires heavy English editing to warrant the manuscript to be published in Sustainability (especially professional proofreading). Figures that are not prepared by authors should have the source (especially icons in Figure 1, Table 3, Table D - or Authors should create their own icons or schemes having in mind copyrights).
The manuscript should be formatted according to the journal guidelines using the Microsoft Word template or LaTex template. According to instructions for Authors references should be at the end of the manuscript - not in footnotes. In this paper the way of citing other authors consists of a mix of different styles.
Finally, the discussions and elaborations about the achieved results are very shallow, there is a need to provide more in-depth details about the changes in development of Islamic banking related to the SDGs before and during the pandemic. The paper requires also clarifcatins about the findings. The reviewer do not see a novelty in this research manuscript.
Author Response
Dear Respected Reviewer 3:
I really appreciate your insightful comments and suggestions. All of the said changes have been addressed accordingly. Kindly refer to the attached (rebuttal/corrections file). Thank you!
Kind Regards:

Round 2
Reviewer 1 Report
Authors have addressed most of my concerns, and they made an important effort to improve their paper. I still have some concerns and suggestions as follows:
Lines 372-374: please, expand the description relative to the Axial Coding Method. Include also references, as not all readers are familiar with this research approach.
Line 375: Step 3: Alignment. I believe that “Alignment” is the most critical step of the process, because of the high implied subjectivity. In what way can a robust alignment be assured? Is there any method that might be used to check such robustness? For instance, performing different rounds of alignment, each time involving different experts, or moving back from results to the step when objective and indicators are still split into two separate groups to test consistency. Please, provide arguments about this issue.
Author Response
Dear Reviewer,
Thank you very much for your logical review. All the said changes are incorporated (refer to the attachment). Thank You!
Kind Regards:
Dr Amin Jan

Reviewer 2 Report
The triple bottom line theory has fundamental theoretical shortcomings.If the Islamic banking system is based on this theory, then there are also problems with its structure.
The scientific value of an article is enhanced by the distance from the research topic.
A good theory also has (may have) weaknesses, and exploring these is an important task for the researcher. I consider the basic clarification of this problem to be a basic condition for publication.
Author Response

(The authors gave the same response as above.)

Reviewer 3 Report
This manuscript version is definitely better than the previous one. It means that the paper has been significantly improved.
I recommend Authors to check the Figure 1 - it is not clear why the lowest raw of this scheme is empty (p. 4).
It is doubt whether table 2 is in fact a form of the table or this chart should be named as a figure (p. 4).
Figure no. 3 should be numbered as Figure 2. The next figures also require changing their numbers.
Thanks to proofreading the content is more lucid and clear for everyone.
Author Response

(The authors gave the same response as above.)
